# MQBench: Towards Reproducible and Deployable Model Quantization Benchmark

**Yuhang Li**[*], **Mingzhu Shen**[*], **Jian Ma**[*], **Yan Ren**[*], **Mingxin Zhao**[*],
**Qi Zhang**[*], **Ruihao Gong**[*], **Fengwei Yu**, **Junjie Yan**
SenseTime Research
http://mqbench.tech/

## Abstract

Model quantization has emerged as an indispensable technique to accelerate deep learning inference. While researchers continue to push the frontier of quantization algorithms, existing quantization work is often unreproducible and undeployable. This is because researchers do not choose consistent training pipelines and ignore the requirements for hardware deployments. In this work, we propose Model Quantization Benchmark (MQBench), a first attempt to evaluate, analyze, and benchmark the reproducibility and deployability for model quantization algorithms. We choose multiple different platforms for real-world deployments, including CPU, GPU, ASIC, DSP, and evaluate extensive state-of-the-art quantization algorithms under a unified training pipeline. MQBench acts like a bridge to connect the algorithm and the hardware. We conduct a comprehensive analysis and find considerable intuitive or counter-intuitive insights. By aligning the training settings, we find existing algorithms have about the same performance on the conventional academic track. While for the hardware-deployable quantization, there is a huge accuracy gap which remains unsettled. Surprisingly, no existing algorithm wins every challenge in MQBench, and we hope this work could inspire future research directions.

## 1   Introduction

Modern deep learning is increasingly consuming larger memory and computation to pursue higher performance. While large-scale models can be trained on the cloud, transition to edge devices during deployment is notoriously hard due to the limited resource budget, including latency, energy and memory consumption. For this reason various techniques have been developed to accelerate the deep learning inference, including model quantization [1, 2, 3, 4, 5], pruning [6, 7, 8, 9, 10], neural network distillation [11, 12], lightweight network design [13], and weight matrix decomposition [14].

In this work, we focus on model quantization for efficient inference. Quantization targets to map the (nearly) continuous 32-bit floating-point (FP) numbers into discrete low-bit integers. As a result, the neural networks could rely on the integer-arithmetic units to speed up the inference. In academic research, there is a trend towards steadily reducing the bit-width and maintaining the accuracy across a range of quantized network architectures on ImageNet. It is incredible that the even 3-bit quantization of both weights and activations can reach FP-level accuracy [15]. Exciting though the breakthrough is, there lacks a systematic study that whether these research works can really be applied to practice, and whether the major improvement is brought by the algorithm rather than the training techniques.

We point out two long-neglected key factors in quantization research, namely reproducibility and deployability. First, we observe that the training hyper-parameters can significantly affect the performance of a quantized network. As an example, Esser *et al*. [15] adopt cosine annealed learning

---

[*]Equal Contributions.

Correspondence to Yuhang Li <liyuhang1@sensetime.com>, Ruihao Gong <gongruihao@sensetime.com>.

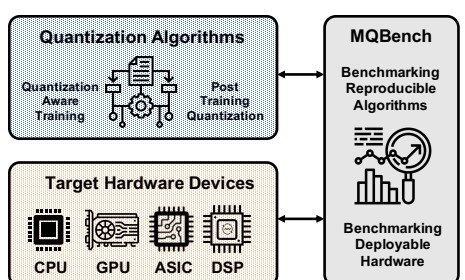

| | Problems | Others | MQBench |
|---|---|---|---|
| **Reproduce** | Unified Hyper-parameters | ✗ | ✓ |
| | Unified Training Pipelines | ✗ | ✓ |
| | Diverse Architectures | ✗ | ✓ |
| | Open Source | ✳ | ✓ |
| **Deploy** | # Supported Hardware | 0 or 1 | 5 |
| | Hardware Quantizer | ✳ | ✓ |
| | Fold BN | ✳ | ✓ |
| | Graph Alignments | ✗ | ✓ |

**Figure 1 & Table 1:** Left: The placement of MQBench, which connects the algorithms and hardware. Right: comparison between MQBench and other quantization works. ✓: condition satisfied, ✗: condition not satisfied, ✳: condition satisfied only in part of existing work.

rate [16] and better weight decay choice, improving the Top-1 accuracy of 2-bit ResNet-18 [17] by 0.7% and 0.4% on ImageNet. Full precision network pre-training can also boost quantization results [15,18]. The reproducibility issue has received considerable attention in other areas as well, e.g. NAS-Bench-101 [19]. So far, there lacks a benchmark that unifies training pipelines and compares the quantization algorithms in a thorough and impartial sense.

Second, we find that the majority of the academic research papers do not test their algorithms on real hardware devices. As a result, the reported performance may not be reliable. For one thing, hardware will fold Batch Normalization (BN) layers [20] into convolutional layers [3] to avoid additional overhead. But most research papers just keep BN layers intact. For another, research paper only considers quantizing the input and weights parameters of the convolutional layers. While in deployment the whole computational graph should be quantized. These rules will inevitably make quantization algorithms less resilient. Another less studied problem is the algorithm robustness: *What will happen if one algorithm is applied to per-channel quantization but it is designed to per-tensor quantization at first?* The algorithm should incorporate the diversity of quantizers design. All these problems suggest a large gap between academic research and real-world deployments.

In this work, we propose **M**odel **Q**uantization **Bench**mark (MQBench), a framework designed to analyze and reproduce quantization algorithms on several real-world hardware environments (See Fig. 1 & Table 1). We carefully studied existing quantization algorithms and hardware deployment settings to set up a bridge between the algorithms and hardware. To complete MQBench, we utilize over 50 GPU years of computation time, in an effort to foster both reproducibility and deployability in quantization research. Meanwhile, our benchmark offers some overlooked observations which may guide further research. To our best knowledge, this is the first work that benchmarks quantization algorithms on multiple general hardware platforms.

In the following context of this paper, we first build a benchmark for reproducing algorithms under unified training settings in Sec. 2. We introduce the requirements for hardware deployable quantization in Sec. 3. Then we conduct extensive experimental evaluation and analysis in Sec. 5. Due to the space limit, we put related work as well as the visualization results in the Appendix.

## 2    MQBench: Towards Reproducible Quantization

In this section, we benchmark the reproducibility of quantization algorithms, mainly including Quantization-Aware Training (QAT)[2]. We evaluate the performance of algorithms on ImageNet [21] classification task. Other tasks like detection, segmentation and language applications are not considered for now since few baseline algorithms were proposed. MQBench evaluation is performed in 4 dimensions: supported inference library given a specific hardware, quantization algorithm, network architecture, and bit-width.

**Hardware-aware Quantizer.**    Throughout the paper, we mainly consider uniform quantization, since the non-uniform quantization requires special hardware design. We use $w$ and $x$ to denote the weight matrix and activation matrix in a neural network. A complete uniform quantization process includes *quantization operation* and *de-quantization operation*, which can be formulated by:

$$\bar{w} = \text{clip}\left(\left\lfloor \frac{w}{s} \right\rceil + z, N_{min}, N_{max}\right), \quad \hat{w} = s \cdot (\bar{w} - z) \tag{1}$$

---

[2]We also build an equally thoroughgoing benchmark for Post-Training Quantization (PTQ) in Appendix. C.

**Table 2:** Comparison of (1) the different hardware we selected and (2) the different QAT algorithms. *Infer. Lib.* is the inference library; *FBN* means whether fold BN. ✳ means undeployable originally, but can be deployable when certain requirements are satisfied.

| Infer. Lib. | Provider | HW Type | Hardware | $s$ Form. | Granularity | Symmetry | Graph | FBN |
|---|---|---|---|---|---|---|---|---|
| TensorRT [22] | NVIDIA | GPU | Tesla T4/P4 | FP32 | Per-channel | Symmetric | 2 | ✓ |
| ACL [23] | HUAWEI | ASIC | Ascend310 | FP32 | Per-channel | Asymmetric | 1 | ✓ |
| TVM [25] | OctoML | CPU | ARM | POT | Per-tensor | Symmetric | 3 | ✓ |
| SNPE [24] | Qualcomm | DSP | Snapdragon | FP32 | Per-tensor | Asymmetric | 3 | ✓ |
| FBGEMM [26] | Facebook | CPU | X86 | FP32 | Per-channel | Asymmetric | 3 | ✓ |

| Algorithms | Deployable | Uniformity | Quant. Type | $s$ Form. | Granularity | Symmetry | Graph | FBN |
|---|---|---|---|---|---|---|---|---|
| LSQ [15] | ✳ | Uniform | learning-based | FP32 | Per-tensor | Symmetric | 1 | ✗ |
| APoT [27] | ✗ | Non-uniform | learning-based | FP32 | Per-tensor | Symmetric | 1 | ✗ |
| QIL [18] | ✗ | Uniform | learning-based | FP32 | Per-tensor | Symmetric | 1 | ✗ |
| DSQ [28] | ✳ | Uniform | rule-based | FP32 | Per-tensor | Symmetric | 1 | ✗ |
| LQ-Net [29] | ✗ | Non-uniform | rule-based | FP32 | Per-tensor | Symmetric | 1 | ✗ |
| PACT [30] | ✳ | Uniform | learning+rule | FP32 | Per-tensor | Symmetric | 1 | ✗ |
| DoReFa [31] | ✳ | Uniform | rule-based | FP32 | Per-tensor | Symmetric | 1 | ✗ |

where $s \in \mathbb{R}_+$ and $z \in \mathbb{Z}$ are called *scale* and *zero-point*, respectively. $\lfloor \cdot \rceil$ rounds the continuous numbers to nearest integers. Eq. (1) first quantizes the weights or activations into target integer range $[N_{min}, N_{max}]$ and then de-quantizes the integers to original range. Given $t$ bits, the range is determined by $[-2^{t-1}, 2^{t-1} - 1]$. We can divide the quantizer based on several metrics: *(1) Symmetric or asymmetric quantization*: For symmetric quantization the zero-point is fixed to 0, while the asymmetric quantization has an adjustable zero-point to adapt different range; *(2) Per-tensor or per-channel quantization*: The per-tensor quantization uses only one set of scale and zero-point for a tensor in one layer while per-channel quantization quantizes each weight kernel independently (i.e. for each row of weight matrix: $\boldsymbol{w}_{i,:}$); *(3) FP32 (32-bit Floating Point) scale or POT (Power of Two) scale*: FP32 scale is nearly continuous, while power-of-two scale is much more challenging. However, POT scale may offer further speed-up.

We select 5 general hardware libraries to evaluate the quantization algorithms, including NVIDIA's TensorRT [22] for Graphics Processing Unit (GPU) inference, HUAWEI's ACL [23] for Application-Specific Integrated Circuit (ASIC) inference, Qualcomm's SNPE [24] for mobile Digital Signal Processor (DSP), TVM [25] for ARM Central Processing Unit (CPU), FBGEMM [26] for X86 server-side CPU. We summarize their implementation details for quantization in Table 2 upper side. Each hardware setting corresponds to a unique quantizer design. Thus, the developed algorithm must be robust to adapt different quantizer configurations. We put the detailed setup for hardware environments in Appendix. E.

**Algorithm.** For quantization-aware training, we compare 6 different algorithms [15, 18, 27, 28, 29, 30, 31]. However, several algorithms cannot be deployable even if we align the quantizer configuration and the other requirements. We put the summary of them in Table 2 lower side. All these algorithms use per-tensor, symmetric settings. We refer this type as *academic setting*. We also identify the quantizer type as learning-based, which learns the scale, or rule-based, which directly computes the scale with heuristics. For a detailed description of these algorithms and the reason why they can be extended to deployable quantization, please see Appendix. F.

**Network Architecture.** We choose ResNet-18 and ResNet-50 [17] as they are most widely used baseline architectures. We also adopt MobileNetV2 [13] which is a lightweight architecture with depthwise separable convolution. In order to quantize EfficientNet [32], we leverage its *Lite* version [33] that excludes the squeeze-and-excitation block and replaces swish activation to ReLU6 for better integer numeric support on hardware. Finally, we add an another advanced architecture RegNetX-600MF [34] with group convolution.

**Bit-width.** In this paper, we mainly experiment with 8-bit post-training quantization (Appendix. C) and 4-bit quantization-aware training. To test the accuracy of the quantized model, we simulate the algorithm with fake quantization (see difference between fake and real quantization in Sec. 3.1). Unlike the reported results in other paper, 4-bit QAT in our benchmark could be very challenging. We do not experiment with 3-bit quantization because it is undeployable on general hardware. As for 2-bit quantization, we find most of the algorithms do not converge on hardware settings.

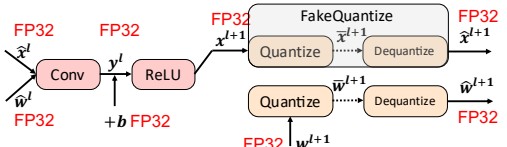
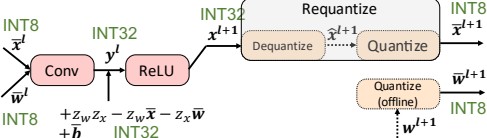

(a) Fake quantization for simulated-training.

(b) Real quantization in deployments.

**Figure 2:** Example computation diagram of a single 8-bit quantized convolutional layer in GPU training or real hardware deployments.

## 2.1 Training Pipelines and Hyper-parameters

Early work like [30, 31] trains the quantized model from scratch, which may have inferior accuracy than fine-tuning [35]. Besides, each paper may have different pre-trained models for initialization. In MQBench, we adopt fine-tuning for all algorithms and each model is initialized by the same pre-trained model, eliminating the inconsistency at initialization.

We adopt standard data prepossessing for training data, including `RandomResizeCrop` to 224 resolution, `RandomHorizontalFlip`, `ColorJitter` with brightness= 0.2, contrast= 0.2, saturation = 0.2, and hue= 0.1. The test data is centered cropped to 224 resolution. We use 0.1 label smoothing in training to add regularization. No other advanced augmentations are further adopted. All models are trained for 100 epochs, with a linear warm-up in the first epoch. The learning rate is decayed by cosine annealing policy [16]. We use the SGD optimizer for training, with 0.9 momentum and Nesterov updates. Other training hyper-parameters can be found in Table 3 aside. We discuss our choice of this set of hyper-parameter in the Appendix. A.

**Table 3:** Training hyper-parameters. *Batch Size* is the batch size per GPU. * means 0 weight decay for BN parameters.

| Model | LR | $L2$ Reg. | Batch Size | # GPU |
|---|---|---|---|---|
| ResNet-18 | 0.004 | $10^{-4}$ | 64 | 8 |
| ResNet-50 | 0.004 | $10^{-4}$ | 16 | 16 |
| EffNet&MbV2 | 0.01 | $10^{-5}*$ | 32 | 16 |
| RegNet | 0.004 | $4 \times 10^{-5}$ | 32 | 16 |

# 3 MQBench: Towards Deployable Quantization

## 3.1 Fake and Real Quantization

Given a weight matrix $\hat{\boldsymbol{w}}$ and an activation matrix $\hat{\boldsymbol{x}}$, the product is given by

$$\boldsymbol{y}_{ij} = \underbrace{\sum_{k=1}^{n} \hat{\boldsymbol{w}}_{ik}\hat{\boldsymbol{x}}_{kj}}_{\text{Fake Quantize}} = s_w s_x \underbrace{\sum_{k=1}^{n} (\bar{\boldsymbol{w}}_{ik}\bar{\boldsymbol{x}}_{kj} - z_w\bar{\boldsymbol{x}}_{kj} - z_x\bar{\boldsymbol{w}}_{ik} + z_w z_x)}_{\text{Real Quantize}}, \tag{2}$$

where $y$ is the convolution output or the pre-activation. In order to perform QAT on GPU, we have to simulate the quantization function with FP32, denoted as the left *Fake Quantize* bracket. For the practical inference acceleration, we have to utilize *integer-arithmetic-only* [3], denoted as the right *Real Quantize* bracket. In Fig. 2, we draw the computational graph of fake quantization and real quantization to reveal their relationship.

For fake quantization, the weights and input activations are quantized and de-quantized before convolution. The intermediate results as well as the bias term, are all simulated with FP32.

As for deployments in real-world, the computation in the *Real Quantize* bracket is integer-only and is accumulated using INT32. One can further optimize the convolution kernels by performing the last two terms offline, since $\boldsymbol{w}$ and $z_w, z_x$ are determined prior to deployment [36]. For bias parameters, we can keep them in INT32, quantized by $\bar{\boldsymbol{b}} = \lfloor \hat{\boldsymbol{b}}/s_b \rfloor$, where $\bar{\boldsymbol{b}}$ is INT32 and $s_b = s_w s_x$. As a result, the bias can be easily fused into Eq. (2). Then, the de-quantization will do the scaling outside the bracket. In the deployment, the de-quantization of the output and the further quantization to integers is fused together, called *Requantization*, given by

$$\hat{\boldsymbol{x}}^{l+1} = s_{w^l} \cdot s_{x^l} \cdot \boldsymbol{x}^{l+1}, \quad \bar{\boldsymbol{x}}^{l+1} = \text{clip}\left(\left\lfloor \frac{\hat{\boldsymbol{x}}^{l+1}}{s_{\hat{x}^{l+1}}} \right\rceil + z_{\hat{x}^{l+1}}, N_{min}, N_{max}\right) \tag{3}$$

We should point out that these two graphs may have some tiny and unavoidable disparity, mainly resulting from the difference between FP32 in simulation and real integers in deployments.

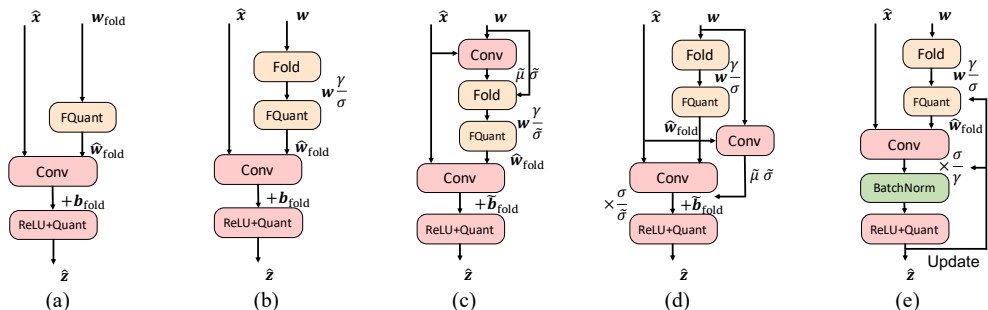

**Figure 3:** Comparison of different Batch Normalization folding technologies. (a) Removing BN layers and directly update $\boldsymbol{w}_{\text{fold}}$. (b) BN folding without any statistics update. (c) BN folding with two convolutions. (d) folding with running statistics and also requires two convolutions. (e) folding running statistics with an explicitly BN layer in training. *Graph (bcde) can be transformed to (a) during inference.* FQuant=FakeQuantize.

## 3.2 Folding Batch Normalization

Batch Normalization (BN) layers are designed to reduce internal covariate shift [20] and also smooth the loss surface [37] for fast convergence. BN introduces a two-step linear transformation for each convolutional layer output. In inference, the linear transformations could be fused into convolutional kernels so that no extra operations are needed, given by:

$$\boldsymbol{w}_{\text{fold}} = \boldsymbol{w}\frac{\gamma}{\sqrt{\sigma^2 + \epsilon}}, \quad \boldsymbol{b}_{\text{fold}} = \beta + (\boldsymbol{b} - \mu)\frac{\gamma}{\sqrt{\sigma^2 + \epsilon}}, \tag{4}$$

where $\mu, \sigma^2$ are the running mean, variance and $\gamma, \beta$ are the affine weight, bias, respectively. $\epsilon$ is for numerical stability (for simplicity, we omit this term in the rest of the paper). If we put quantization after the folding of BN layers, there will be no extra floating-point operations during inference. However, BN folding does not draw much attention in existing literature and causes deployment issues. In this section, we will discuss 5 possible strategies for BN folding. We denote the current batch mean and variance as $\tilde{\mu}, \tilde{\sigma}^2$. The diagram of these 5 types are demonstrated in Fig. 3.

**Strategy 0:** Merge the parameters into weights and bias with Eq. (4), and remove this layer entirely (Fig. 3(a)). We find this choice cannot be trained with large learning rate because of the gradient explosion without BN. Consequently, extensive hyper-parameter searching is necessary.

**Strategy 1:** Fold BN layers and do not update the running statistics (Fig. 3(b)). Nevertheless, the affine parameters $\gamma, \beta$ can still be updated with SGD. We find this strategy can still smooth the loss landscape and achieve comparable accuracy even no statistics are updated. This folding strategy also significantly reduces the training time by avoiding statistics synchronization.

**Strategy 2:** Introduced in [3], this folding strategy can update running statistics (Fig. 3(c)). The convolution will be calculated twice during training, which causes additional overhead. The first time is to compute the batch mean and variance $\tilde{\mu}, \tilde{\sigma}^2$ using FP32 weights. Then, the current batch mean, variance are folded into weight kernels. During inference, the weights and biases will be folded using running mean and variance as in Eq. (4).

**Strategy 3:** Introduced in [1], this option also calculates the convolution twice. The first time is the same with strategy 2, which will estimate $\tilde{\mu}, \tilde{\sigma}^2$. However, in this strategy the weights will be folded with *running statistics* to avoid the undesired fluctuations of the *batch statistics*. The batch variance factor will be used to re-scale the output after the second convolution, as shown in Fig. 3(d).

**Strategy 4:** Introduced in PyTorch quantization repo [38], this option does not cost two times convolution but explicitly adds BN layers after the quantized convolution. One of the benefits brought by this strategy is that batch statistics are calculated based on quantized weight. During inference, the re-scaling of output $\frac{\sigma}{\gamma}$ can be neutralized by BN, therefore the graph can be transformed to Fig. 3(a).

## 3.3 Block Graph

Most academic papers only consider quantizing the input and the weight kernels of convolutional or fully connected layers. However, modern neural architectures includes other operations, like

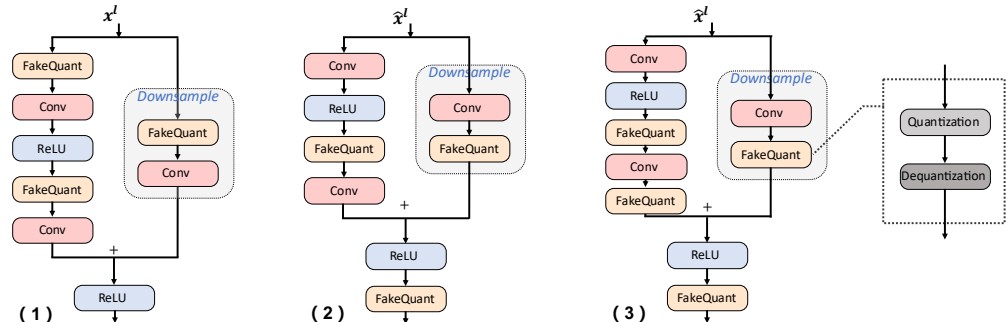

**Figure 4:** Comparison of different quantization implementations for a basic block in the ResNet [17].

elementwise-add in ResNet [17] and concatenation in InceptionV3 [39]. In addition, different hardware will consider different levels of graph optimization and can propose different solutions to construct a graph for quantized neural networks. In MQBench, we sort out different implementations and summarize them in a schematic diagram (Fig. 4). Note that Fig. 4 only gives an example of a basic block in ResNet-18/-34. The bottleneck block in ResNet-50 can be naturally derived from this diagram. We also put the diagram of the inverted residual bottleneck of MobileNetV2 [13], and concatenation quantization in the Appendix. G.

Fig. 4 left shows the conventional academic implementations for quantizing a basic block. Only the input of convolutional layers will be quantized to 8-bit. (Note that in academic papers the INT8 means both quantization and de-quantization.) The block input and output as well as the elementwise-add all operate at full precision. Consequently, the network throughput hasn't been reduced, and will significantly affect the latency. In some architectures, this graph even won't bring any acceleration since the latency is dominated by I/O. Another problem is the separate quantization of the activation in the downsample block, which also brings undesired costs.

Fig. 4 middle presents the NVIDIA's TensorRT [22] implementation for basic block. We can find that the input and output must be quantized to low-bit to reduce the data throughput. Low bit input can ensure two branches will use the same quantized activation in the downsample block. As for the elementwise-add layer, it will be conducted in a 32-bit mode due to the fusion with one of the former convolutional layer's bias addition. Thus only one of its inputs will be quantized. Fig. 4 right demonstrates the implementation in other hardware library, such as FBGEMM [26] and TVM [25]. The only difference is that they require all inputs of the elementwise-add to be quantized. In 4-bit symmetric quantization, this can severely affect the accuracy.

## 4   MQBench Implementation

We implement MQBench with Pytorch [40] package, with the support of the latest feature, the `torch.fx` (also known as FX, see documentation in [41]) in version 1.8. FX contains a symbolic tracer, an intermediate representation, and Python code generation, which allows for deeper meta programming. We implement the quantization algorithms and the hardware-aware configurations in MQBench. Only an API call is needed to trace a full precision model and convert it to a quantized model. A code demo of quantizing ResNet-18 with TensorRT backend is presented in , For more details, readers are recommended to see the official repository.

## 5   MQBench Evaluation

In this section, we conduct a thorough evaluation and analysis for quantization algorithms, network architectures, and hardware. We study several evaluation metrics, given by ① **test accuracy:** the Top-1 accuracy on the ImageNet validation set, it directly reflects the task performance of the algorithm; ② **hardware gap**: the difference between hardware and academic test accuracy, this metric can reflect the impact of deployable quantization on the algorithm; ③ **hardware robustness**: the average test accuracy on 5 hardware environments. ④ **architecture robustness**: the average test accuracy on 5 network architectures. These last two metrics are often neglected by most of the existing literature but

**Algorithm 1** Training and depolying quantized model with MQBench

```python
import torchvision.models as models
from mqbench.convert_deploy import convert_deploy
from mqbench.prepare_by_platform import prepare_by_platform, BackendType
from mqbench.utils.state import enable_calibration, enable_quantization

# first, initialize the FP32 model with pretrained parameters.
model = models.__dict__["resnet18"](pretrained=True)

# then, we will trace the original model using torch.fx and \
# insert fake quantize nodes according to different hardware backend (e.g. TensorRT).
model = prepare_by_platform(model, BackendType.Tensorrt)

# before training, we recommend to enable observers for calibration and then enable quantization.
model.eval()
enable_calibration(model)
# calibrate
for i, batch in enumerate(calibration_data):
    # do forward procedures
    ...
enable_quantization(model)

# training loop
model.train()
for i, batch in enumerate(data):
    # do forward and backward procedures
    ...

# deploy model, remove fake quantize nodes and dump quantization params like clip ranges.
convert_deploy(model.eval(), BackendType.Tensorrt, input_shape_dict={'data': [10, 3, 224, 224]})
```

may have a significant value. In the Appendix. B, we include more study and provide the diagnostic information for the quantization benchmark. **Results are from MQBench *V0.0.1*.**

## 5.1 Evaluation with Academic Setting.

**Table 4:** Academic setting benchmark for 4-bit quantization-aware training, result in bracket is the reported accuracy in original paper. "NC" denotes not converged.

| Model | LSQ [15] | APoT [27] | QIL [18] | DSQ [28] | PACT [30] | DoReFa [31] |
|---|---|---|---|---|---|---|
| ResNet-18 | 70.7 (71.1) | 70.5 (70.7) | **70.8** (70.1) | 70.0 (69.6) | 70.5 (69.2) | 70.7 (68.1) |
| ResNet-50 | **77.4** (76.7) | 77.1 (76.6) | 77.2 (N/A) | 76.4 (N/A) | 76.3 (76.5) | 76.4 (71.4) |
| MobileNetV2 | 70.6 (66.3) | 68.6 (N/A) | 70.3 (N/A) | 69.6 (64.8) | **70.7** (61.4) | NC (N/A) |
| EfficientNet-Lite0 | 72.6 (N/A) | 70.0 (N/A) | 72.7 (N/A) | 72.6 (N/A) | **73.0** (N/A) | NC (N/A) |
| RegNetX-600MF | 72.7 (N/A) | **73.0** (N/A) | 71.6 (N/A) | 71.7 (N/A) | 72.2 (N/A) | 72.9 (N/A) |
| Avg. Arch. | **72.8** | 71.9 | 72.5 | 72.1 | 72.5 | 44.0 |

We first revisit the performance of quantization in academic settings (per-tensor, symmetric quantization without any bn folding, etc.). This setting is predominately adopted in the research paper. We summarize our benchmark results and the originally reported results (in bracket) in Table 4. By aligning the training hyper-parameters and pipeline, we find several surprising results.

① **The difference between algorithms is not as significant as reported in the original paper.** As can be seen, the maximum Top-1 accuracy difference of ResNet-18 is 0.8% (= QIL − DSQ), which is much smaller than 3.0% as compared in the original paper. This phenomenon is even more evident in the case of ResNet-50, where the maximum difference as reported is 5.3% while the actual maximum difference is only 1.1%. This suggests that *80%($^{4.2}/_{5.3}$) of the accuracy improvement from DoReFa to LSQ is made from better training techniques, only 20% comes from the superiority of the algorithm.* On MobileNetV2, prior work reported PACT and DSQ with only 61.4%, 64.8% accuracy, however, our setting can obtain 70.7%, 69.6% candidates respectively, indicating the importance of the unified training hyper-parameters.

② **No algorithm achieves the best or the worst performance on every architecture.** Among 5 different network architectures, there are 4 different winner solutions and 4 different worst solutions. Such an outcome demonstrates that existing algorithms cannot adapt every network architecture very well. We encourage studying the architecture robustness, which is the mean accuracy across architectures. In that case, LSQ achieves the highest robustness. However, the improvement in robustness is also not as evident as we expected before.

③ **The rule-based algorithms can achieve comparable performance with learning-based algorithms.** DoReFa-Net [31], which simply clips the activation to [0, 1], reaches the same 70.7% test

accuracy as LSQ [15] on ResNet-18. It also surpasses the PACT by 0.2%, revealing that even a handcrafted fixed clipping range with the right training pipelines can have state-of-the-art accuracy. Although, DoReFa fails to quantize depthwise conv networks, e.g. MobileNetV2. We believe this is due to the activation range in those networks are much larger than ResNet-family (as can be shown in our diagnostic information in Appendix. B). Nevertheless, we believe rule-based quantization can achieve better performance if a good range can be found in advance.

## 5.2 Evaluation with Graph Implementation

In this section, we study the effect of different computation graphs for quantization networks and algorithms. Graph 1,2,3 correspond to graph implementations in Fig. 4(a), (b), (c). Following BN folding experiments, we modify the *academic* settings and only change the graph implementations. PACT and LSQ are selected for this study, conducted on ResNet-18 and MobileNetV2. The results in Table 5 show the final performance. We find: **unlike BN folding, which is sensitive to algorithms, the graph implementation is sensitive**

**Table 5:** Comparison of the accuracy on 4-bit QAT models, given different graphs implementations.

| Model | ResNet-18 | | | MobileNetV2 | | |
|---|---|---|---|---|---|---|
| Graph | 1 | 2 | 3 | 1 | 2 | 3 |
| LSQ [15] | 70.7 | 70.7 | 70.3 | 70.6 | 67.5 | 67.0 |
| PACT [30] | 70.5 | 70.3 | 69.4 | 70.7 | 68.3 | 67.8 |

**to the network architecture**. For instance, PACT only drops 1.1% accuracy on ResNet-18 by switching graph from 1 to 3. However, the gap can increase to 2.9% on MobileNetV2. The same trend is also observed for LSQ, where 0.4% and 3.6% accuracy degradation are observed on ResNet-18 and MobileNetV2, respectively.

## 5.3 Evaluation with BN Folding

We then study the BN folding strategies designed for QAT. We choose LSQ [15] and PACT [30], running on ResNet-18 and MobileNetV2 for ablation study. Here we do not employ any real hardware-aware quantizer but only modify the conventional *academic* settings (i.e. per-tensor, symmetric) to accommodate BN folding. The results are summarized in Table 6. During our experiments, we have the following observations:

① **BN folding is sensitive to quantization algorithms, and strategy 4 works best generally.** We first examine the LSQ with BN folding, where we find the algorithm converges to a similar performance with normal BN QAT, on both ResNet-18 and MobileNetV2. Unlike LSQ, BN folding has a severe impact on PACT quantized models. All folding strategies except 4 fail to converge on ResNet-18. Even using strategy 4 will decrease 2.7% accuracy. For MobileNetV2, the decrease is more significant (9.9%).

② **Strategy 4 does not obtain any significant speed-up than strategy 2, 3 even it only computes one-time convolution.** Although strategy 4 is faster than 2,3 in forward computation as it has less computation, but it is much slower in gradient calculation. On ResNet-18 LSQ, we find strategy 4 costs 80% more time than 2,3 to do backpropagation.

③ **Updating batch statistics may not be necessary for BN-folding-friendly algorithms like LSQ.** As an example, using strategy 1 in LSQ only drops 0.2%~0.3% accuracy than those who update the batch mean and variance. Moreover, strategy 1 can be 30%~50% faster than them since it does not need to compute or synchronize statistics and has much faster backpropagation.

④ **Synchronization of BN statistics in data-parallel distributed learning can improve accuracy.** In distributed training with normal BN, asynchronous BN statistics across devices are acceptable and will not affect the final performance as long as the batch size is relatively large. However, in QAT folding BN into weights with asynchronous statistics will produce different quantized weights, further magnifying the training instability. Synchronization needs time,[3] therefore it is an accuracy-speed tradeoff. SyncBN can improve 2.3% accuracy for PACT ResNet-18.

⑤ **Folding BN will incur severe instability in the initial training stage and this can be alleviated effectively by learning rate *warm-up*.** Without warm-up, most QAT with BN folding will fail to converge. Therefore, for all the rest experiments which require BN folding, we employ 1 epoch learning rate linear warm-up, synchronized BN statistics, and strategy 4 to facilitate it.

---

[3]The communication costs of synchronization may depend on the equipment of the cluster or the server.

**Table 6:** Comparison of the accuracy on a 4-bit quantized ResNet-18 and MobileNetV2, using LSQ [15] and PACT [30], given different folding strategies ("-1" denotes normal BN training without folding, others are folding strategies introduced in Sec. 3.2.); "NC" denotes Not Converged; "*" denotes asynchronous statistics.

| Model | ResNet-18 | | | | | | | MobileNetV2 | | | | | | |
|---|---|---|---|---|---|---|---|---|---|---|---|---|---|---|
| Folding Strategy | -1 | 0 | 1 | 2 | 3 | 4 | 4* | -1 | 0 | 1 | 2 | 3 | 4 | 4* |
| LSQ | 70.7 | 69.8 | 70.1 | 70.2 | 70.3 | **70.4** | 70.1 | 70.6 | 69.5 | 69.9 | 70.0 | **70.1** | 70.1 | 64.8 |
| PACT | 70.5 | NC | NC | NC | NC | **67.8** | 65.5 | 70.7 | NC | NC | NC | NC | **60.8** | NC |

**Table 7: 4-bit** Quantization-Aware Training benchmark on the ImageNet dataset, given different algorithms, hardware inference libraries, and architectures. "NC" means not converged. Red and Green numbers denotes the decrease and increase of the hardware deployable quantization.

| Model | Method | Paper Acc. | Academic | TensorRT | ACL | TVM | SNPE | FBGEMM | Avg. HW |
|---|---|---|---|---|---|---|---|---|---|
| ResNet-18 FP: 71.0 | LSQ [15] | 71.1 / 70.7[1] | **70.7** | 69.3(1.4) | 70.2(0.5) | 67.7(3.0) | **69.7(1.0)** | **69.8**(0.9) | 69.3±0.87 |
| | DSQ [28] | 69.6 | 70.0 | 66.9(3.1) | 69.7(0.3) | 67.1(2.9) | 68.9(1.1) | 68.9(1.1) | 68.3±1.10 |
| | PACT [30] | 69.2 | 70.5 | 69.1(1.4) | **70.4(0.1)** | 57.5(13.0) | 69.3(1.2) | 69.7(**0.8**) | 67.2±4.87 |
| | DoReFa [31] | 68.1[2] | **70.7** | **69.6**(1.1) | 70.4(0.3) | **68.2(2.5)** | 68.9(1.8) | 69.7(1.0) | **69.4±0.75** |
| ResNet-50 FP: 77.0 | LSQ [15] | 76.7 | **77.4** | **76.3**(1.1) | **76.5**(0.9) | **75.9(1.5)** | **76.2**(1.2) | 76.4(1.0) | **76.3±0.21** |
| | DSQ [28] | N/A | 76.4 | 74.8(1.6) | 76.2(0.2) | 74.4(2.0) | 75.9(**0.5**) | 76.0(0.4) | 75.5±0.72 |
| | PACT [30] | 76.5 | 76.3 | **76.3(0.0)** | 76.1(0.2) | NC | NC | **76.6(0.3)** | 45.8±37.4 |
| | DoReFa [31] | 71.4[2] | 76.4 | 76.2(0.2) | 76.3(**0.1**) | NC | NC | 75.9(0.5) | 45.7±37.3 |
| MobileNetV2 FP: 72.6 | LSQ [15] | 66.3[3] | 70.6 | 66.1(4.5) | 68.1(2.5) | **64.5(6.1)** | **66.3(4.3)** | 65.5(5.1) | **66.1±1.18** |
| | DSQ [28] | 64.8 | 69.6 | 48.4(21.2) | 68.3(1.3) | 29.4(39.8) | 41.3(28.3) | 50.7(18.9) | 47.6±12.7 |
| | PACT [30] | 61.4[4] | **70.7** | **66.5(4.2)** | **70.3(0.4)** | 48.1(22.6) | 60.3(10.4) | **66.5(4.2)** | 62.3±7.8 |
| | DoReFa [31] | N/A | NC | NC | NC | NC | NC | NC | 0±0 |
| EfficientNet-Lite0 FP: 75.3 | LSQ [15] | N/A | 72.6 | 67.0(5.6) | 65.5(7.1) | **65.0(7.6)** | **68.6(4.0)** | 66.9(6.7) | **66.6±1.27** |
| | DSQ [28] | N/A | 72.6 | 35.1(37.5) | 69.6(3.0) | NC | 7.5(65.1) | 45.9(26.7) | 31.6±25.5 |
| | PACT [30] | N/A | **73.0** | **68.2(4.8)** | **72.6(0.4)** | 45.9(27.1) | 56.5(16.5) | **69.0(4.0)** | 62.4±9.88 |
| | DoReFa [31] | N/A | NC | NC | NC | NC | NC | NC | 0±0 |
| RegNetX-600MF FP: 73.7 | LSQ [15] | N/A | 72.7 | **72.5(0.2)** | 72.8(0.1) | **70.0(2.7)** | **72.5(0.2)** | 72.5(0.2) | **72.1±1.04** |
| | DSQ [28] | N/A | 71.7 | 68.6(2.1) | 71.4(0.3) | 64.5(7.2) | 70.0(1.7) | 70.0(1.7) | 68.9±2.37 |
| | PACT [30] | N/A | 72.2 | 72.0(**0.2**) | **73.3(1.1)** | NC | NC | **72.5(0.3)** | 43.6±35.5 |
| | DoReFa [31] | N/A | **72.9** | 72.4(0.5) | 73.2(0.3) | NC | NC | 72.2(0.7) | 43.6±35.6 |

[1,2,3,4] Accuracy reported in [30, 42, 43, 44], respectively.

## 5.4 4-bit QAT

In this section we establish a major baseline for existing and future work by using our deployable and reproducible benchmark to compare some popular algorithms. We experiment with 4-bit quantization-aware training. Unlike academic settings in Table 4 where 4-bit quantization is near-lossless, we showcase the challenging nature of deployable quantization. Like [19], our intention is not to provide a definite answer to *"Which methods work best on this benchmark?"*, but rather to demonstrate the utility of a reproducible and deployable baseline. And hopefully, with newly discovered insights, we can guide the future study on quantization algorithms. The major results are presented in Table 7.

**Test Accuracy.** We apply the algorithm to 5 distinct real-world hardware deployment environments and reported their *fake quantization performance*. We first visit the absolute test accuracy. As can be seen from the table, we observe **no algorithms achieve the best absolute performance on every setting.** Among the 25 settings (5 architectures × 5 hardware environments), LSQ [15] obtains the best performance in 52% cases.[4] PACT [30] and DoReFa [31] attain the rest 38% and 10% best practices. Although DSQ [28] does not achieve any best practice, we should never rank it to the last one. In many cases, DoReFa and PACT do fail to converge, while DSQ can have a good performance. We cannot simply rank each algorithm based on a single metric. We also study the distribution of the test accuracy. In Fig. 5, we show the **standard deviation of the test accuracy** with different combinations of hardware and network architecture. This metric measures the performance difference between algorithms. Lower variance means less distinction between algorithms. Based on Fig. 5, we find depthwise conv-net (MobileNetV2

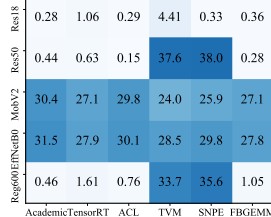

**Figure 5:** Variance of test accuracy.

---

[4]We compute the best solution count in half if the algorithm is tied for the first place.

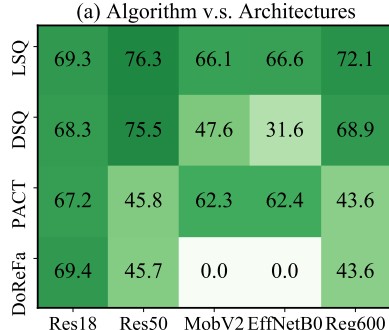 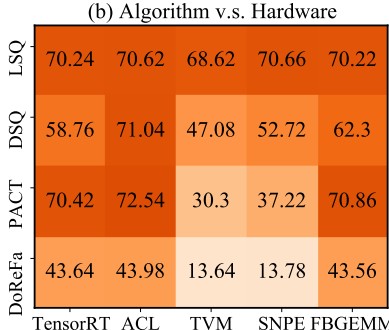

**Figure 6:** Measuring the mean accuracy of algorithms on network architectures (left) and hardware (right).

and EfficientNet-Lite) and per-tensor quantization (TVM and SNPE) have large variance. Thus we recommend paying more attention to them in the future study.

**Hardware Gaps.** We also investigate the hardware gap metric, which means the degradation when transiting from academic setting to hardware setting. The values are marked with colored numbers in the table. Notably, **93% of the experiments encounter accuracy drop**. Among them, 25.8% settings drop within 1.0% accuracy, 47.3% settings drop within 3.0% accuracy. Similar to our findings in absolute accuracy, **no algorithms achieve the least hardware gap in every setting.** LSQ only has a 36% probability to win the hardware gap metric while PACT has a probability of 48%.

**Hardware & Architecture Robustness.** We verify the architecture and hardware robustness in Fig. 6. For architecture robustness, we discover three types of patterns. On ResNet-18, each algorithm can converge to high accuracy, while ResNet-50 and RegNet share a different pattern with ResNet-18. Finally, MobileNetV2 and EfficientNet-Lite also have similar algorithm performances. This suggests that networks architectures can have different sensitivity to quantization algorithms.

We also explore the robustness of quantization algorithms when they are applied to various hardware. Generally, LSQ has the best hardware robustness. It brings much more stable performance on different hardware. However, we find LSQ is not suitable for per-channel quantization. For all 15 per-channel hardware settings, LSQ only wins 23% cases, while 66.7% of the trophy is claimed by PACT. LSQ exhibits exciting superiority in per-tensor quantization, where it wins 90% cases. This result indicates the importance of the hardware robustness metric.

**Suggestions for Algorithm Selection.** Although no single algorithm is SOTA for all cases in QAT, there are some underlying rules for algorithm selection. Based on Fig. 6, we find generally LSQ performs best in per-tensor quantization (SNPE and TVM), while PACT performs best in per-channel quantization (TensorRT, ACL, FBGEMM). Therefore, we recommend using PACT for per-channel quantization and LSQ for per-tensor quantization. If the target hardware or the network architectures are not met before, we recommend using LSQ since it has the best average performance in history (Fig. 6). Note that not all cases need careful algorithm selection. According to Fig. 5, in the case of per-channel quantization and non-depthwise convolution network, the variance of algorithm is quite low. Therefore, in these settings, the selection of algorithm is trivial.

# 6 Discussion

In this work we have introduced MQBench, a systematic tabular study for quantization algorithms and hardware. To foster reproducibility, we align the training hyper-parameters and pipelines for quantization algorithms. To foster deployability, we sort out 5 hardware deployments settings and benchmark the algorithms on them across 5 network architectures. We conduct a thorough evaluation, focusing on the less explored aspects of model quantization, including BN folding, graph implementations, hardware-aware quantizer, etc. However, MQBench also has limitations, quantization faces more challenges in deployments like object detection and NLP application. More advanced algorithms (like data-free quantization [45, 46, 47]) are also needed for a complete benchmark. These aspects should be studied in the future work. Be that as it may, we hope MQBench will be the first of a continually improving sequence of rigorous benchmarks for the quantization field.

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

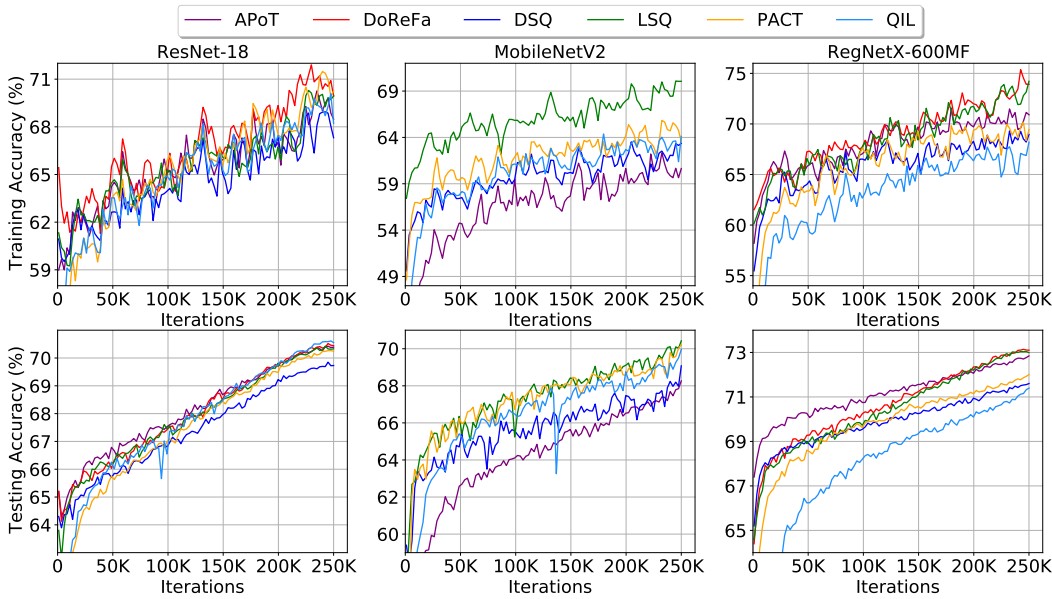

**Figure 7:** Visualization of training and testing accuracy for 6 different algorithms with academic setting on ResNet-18, MobileNetV2 and RegNetX-600MF. Note that DoReFa failed to converge in MobileNetV2 training.

# A    Choice of Hyper-parameters.

We utilize a single, fixed set of training hyper-parameters for all MQBench experiments. As we demonstrated in the experimental evaluation, the hyper-parameters and other training techniques can have a profound impact on the final performance. Thus we have to carefully select the optimal hyper-parameters.

**Optimizer:** In our preliminary experiments, we compare SGD and Adam optimizer. The results are somewhat counter-intuitive. The ranking of Adam and SGD is not consistent in all the experiments. Adam occasionally outperforms SGD in LSQ experiments. While for PACT, DoReFa quantization, SGD tends to have higher performances. Therefore we choose to use SGD for final optimizer. We suggest to study the optimizer's impact on quantization algorithm in future work.

**Learning rate and scheduler:** Following LSQ [15], we use the cosine learning rate decay [16] to perform the quantization-aware training. This scheduler is also adopted in other areas for training the model [19] and generally has good performance. For the initial learning rate, we run a rough grid search in $\{0.04, 0.01, 0.004, 0.001\}$. For ResNet-family (including ResNet-18, -50, RegNetX), we find 0.004 works best. For MobileNet-family (including MobileNetV2 and EfficientNet-Lite0), we find 0.01 works best.

**Weight Decay ($L2$ Regularization):** The weight decay choice also has great impact on the final performance of the quantization algorithms. Following existing state-of-the-art practice [15], 4-bit model can reach FP-level accuracy. Thus a same degree of regularization is appropriate. We find this intuition works well in ResNet-family with academic setting. Therefore, we set the weight decay the same as the full precision training for ResNet-family. For MobileNetV2, we find further decreasing the weight decay from $4e-5$ to $1e-5$ can improve the accuracy, therefore we choose to use this setting. However, we should point out that in hardware-aware settings, the 4-bit QAT cannot reach original FP-level performance, thus tuning the weight decay in hardware setting can further boost the performance. We do not explore the weight decay tuning for each setting in an effort to keep the training hyper-parameters unified.

**Batch size and training iterations:** We train each model for 100 epochs. This is broadly adopted in existing quantization literature. However, for MobileNet-family we find 150 epochs training can further improve the accuracy. We do not employ this choice since 150 epochs training is also time-consuming. As for the batch size choice, we think large batch training can reduce training time but will increase the cost to reproduce the results. Therefore our principle is to limit the maximum number of devices to 16 and to increase the batch size as large as possible. For ResNet-18, we only employ 8 GPUs.

**Clarification:** The hyper-parameters choice is based on the experience in prior works and our simple preliminary exploration. We search the hyper-parameters in academical setting, although we are aware of that the hardware settings can have a different optimal choice. Our intention is to build a benchmark with unified training settings and to eliminate the bias brought by training hyper-parameters.

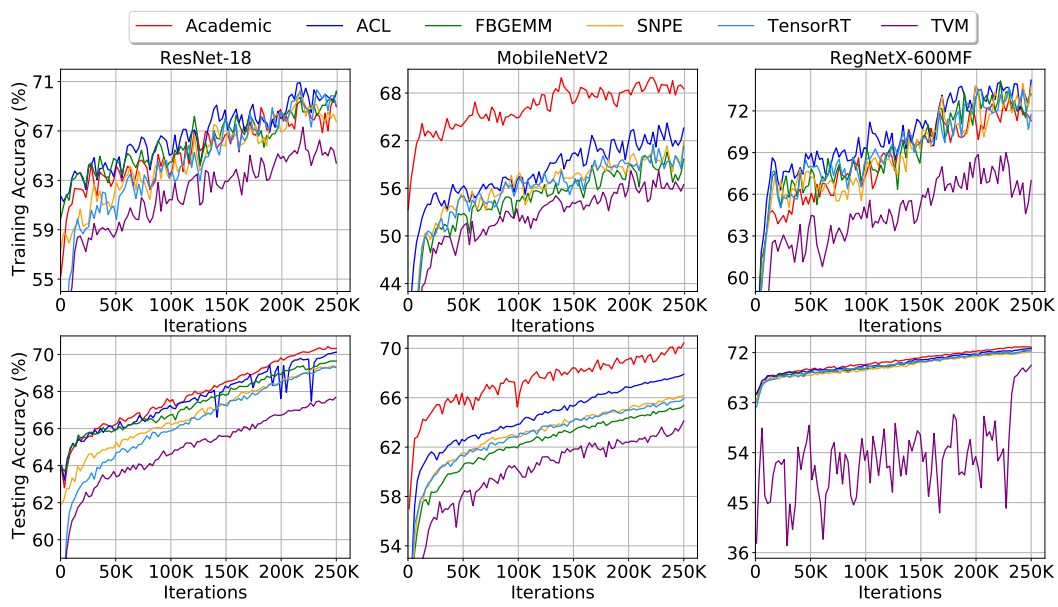

**Figure 8:** Visualization of training and testing accuracy for 6 different setting using LSQ on ResNet-18, MobileNetV2 and RegNetX-600MF.

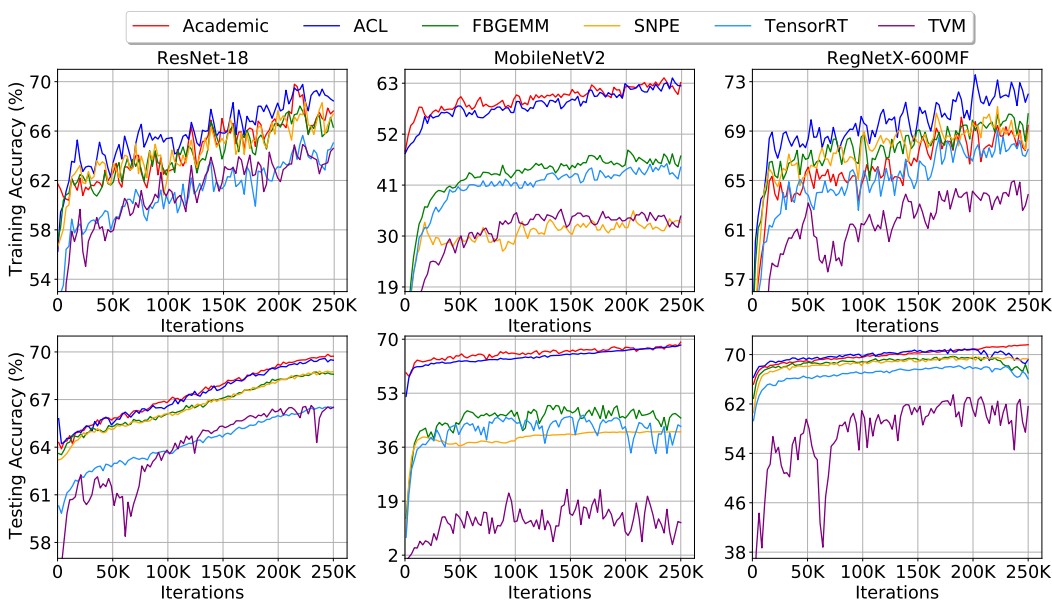

**Figure 9:** Visualization of training and testing accuracy for 6 different setting using DSQ on ResNet-18, MobileNetV2 and RegNetX-600MF.

# B Diagnostic Information

The final test accuracy may be too sparse to evaluate a algorithm or a strategy. In MQBench, we also release the diagnostic information like [48], in order to provide more useful information in studying the quantization. Our diagnostic information includes ①: the training log file and training & testing curve, which reveals the convergence speed and ②: the final checkpoints as well as the quantization parameters, i.e. scale and zero point, which can be used to observe the quantization range.

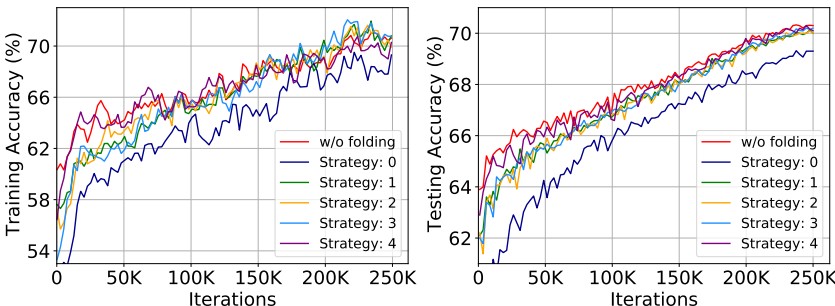

**Figure 10:** Visualization of training and testing accuracy of ResNet18 for 6 different folding BN strategies.

## B.1 Training Curve

In this section we visualize the training curve as well as the test curve in the quantization-aware training. For training accuracy we record the training accuracy per 1k iterations. Then we use exponential moving average with momentum 0.3 to draw the evolution of the training accuracy. For test curve we directly record the validation accuracy in every 1k iterations.

**Algorithm with Academic Setting**. In Fig. 7, we visualize 6 algorithms (LSQ, PACT, DoReFa, QIL, DSQ, APoT) on three architectures, including ResNet-18, MobileNetV2 and RegNetX-600MF. All these quantization algorithms are trained with academic setting. Generally, QIL and PACT has the relatively low initialization accuracy. In terms of convergence speed, we find LSQ, DoReFa perform well. In MobileNetV2 curve, we observe the increasing instability of the test accuracy. QIL even drops 5% accuracy occasionally.

**Hardware**. We visualize the LSQ and DSQ algorithms under 6 different setting, as shown in Fig. 8 and Fig. 9. In a nutshell, we find that ACL (blue) curve has the closest distance with the academic (red) curve. TVM has the lowest curve. This is because the scale in TVM is power-of-two, and the quantizer is per-tensor. On RegNetX-600MF, we observe significant fluctuations of LSQ TVM, indicating the challenge of the real world hardware.

**Folding BN Strategy**. In Fig. 10, we compare different folding BN strategies on ResNet-18 with LSQ algorithm. The strategy 4 has a similar convergence route with regular BN training. However we observe no obvious difference between 1-4 for final convergence. Strategy 0, on the contrary, has the lowest performance.

## B.2 Quantization Range

The quantization range, also called *the clipping range* is essential to quantization performance. The quantization range must be large enough to cover the majority of the activation (clipping error), at the same time, it should be small enough to ensure the rounding error won't get too large. In this section we visualize the activation quantization range (weights quantization range is less informative because some algorithms transform the weights to different distribution) under different architectures, algorithms, and hardware settings. This visualization might help to inspire future research in developing the quantization algorithms.

### B.2.1 Models

In this subsection we include the visualization of activation distribution in academic setting. We use LSQ, on 5 different architectures: ResNet-family including (ResNet-18, ResNet-50, RegNetX-600MF), MobileNet-family including (MobileNetV2, EfficientNet-Lite0). The results are visualized in Fig. 11. We should point out that the first layer is the input image and the last layer is the input to the final fully-connected layers, which may have larger range and are quantized to 8-bit. By excluding the first and the last layer we find that the most layers in ResNet-family have less than 1 range. Moreover, in certain layers, the activation has an extremely small range, e.g. (0, 0.1). Such distribution may be lossless if quantization is applied to it. Next, we find the activation in MobileNet-family has much more larger variance. For example, the activation in EfficientNet-Lite0 can range from $-10$ to $+10$. We think two factors contribute to this abnormal distribution. First, the depthwise convolution prevents the communication between channels. As a result, each channel will learn its own range. Second, the shortcut-add and the linear output layer of the block is easy to accumulate the activations across blocks. The formulation is given by

$$F(\boldsymbol{x}) = f(\boldsymbol{x}) + \boldsymbol{x}, \tag{5}$$

We can see that the block output contains the input, so several blocks' output are accumulated. In ResNet-18, the block output will be activated by ReLU, thus restricting the range of the activation.

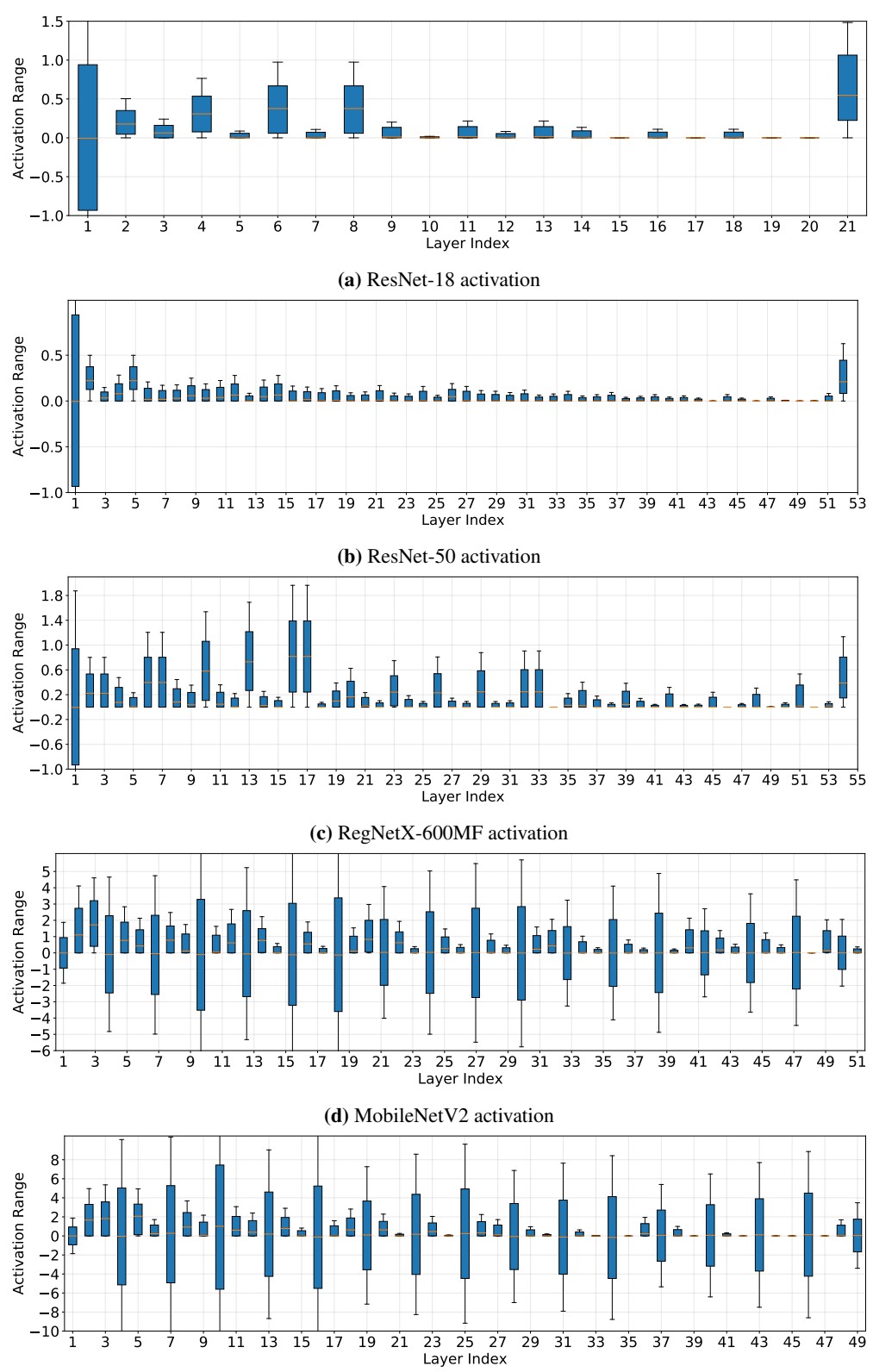

**(a)** ResNet-18 activation

**(b)** ResNet-50 activation

**(c)** RegNetX-600MF activation

**(d)** MobileNetV2 activation

**(e)** EfficientNet-Lite0 activation

**Figure 11:** The activation distribution of LSQ in academic setting.

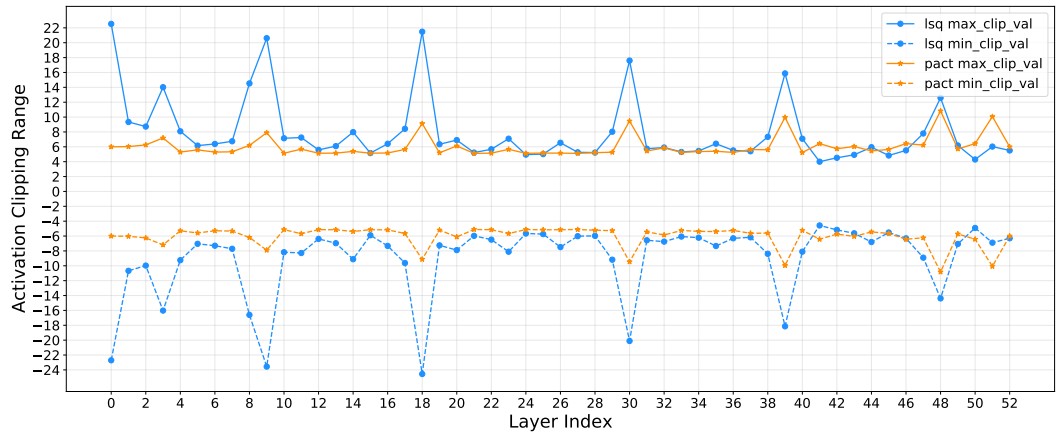

**Figure 12:** Activation clipping range on MobileNetV2 with TensorRT setting.

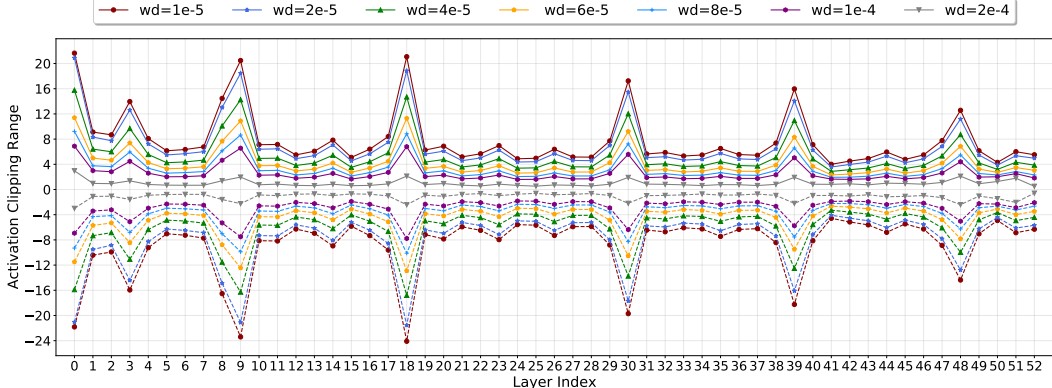

**Figure 13:** Activation clipping range learned by LSQ, given different $L2$ regularization of the scale parameter.

### B.2.2 Algorithm

In this subsection we visualize the activation range on MobileNetV2 with PACT and LSQ. As shown in Fig. 12, the LSQ learns a significant larger clipping range than PACT. For PACT experiments, the maximum clipping range is $[-10, 10]$, however in LSQ the range could be up to $[-20, 20]$. We conjecture this is due to the weight decay. For LSQ optimization, the learnable parameter is the scale, while for PACT optimization the parameter is the clipping threshold. Note that $\alpha = N_{max} \times s$, therefore the regularization effect of $\alpha$ is much more evident than $s$. This phenomenon also proves that LSQ gradients cannot prevent the scale from growing too large.

### B.2.3 Ablation Study

Having observed the large clipping range learned by LSQ, we conduct an ablation study: *imposing different weight decay on the scale parameter*. For other weights and bias parameters we remain the original choice. We run our ablation study on MobileNetV2 with TensorRT setting. The weight decay are chose from $\{1e-5, 2e-5, 4e-5, 6e-5, 8e-5, 1e-4, 2e-4\}$. The results are presented in Table 8 and the clipping range of activation are visualized in Fig. 13. It is easy to observe that the clipping range continues to narrow along with the increasing weight decay. Moreover, we find changing the regularization of scale leads to proportional change, a very similar phenomenon with rule-based clipping range. In terms of final accuracy, weight decay $8e-5$ reaches the highest results and is 2.3% higher than our baseline. We think this result indicates that learning-based range tends to clip less activation, and $L2$ regularization plays an important role to balance the total error.

### B.3 Latency

The final objective of quantization is to accelerate the inference of neural networks. To show the practical efficiency of quantization, we profile six different hardware, including Tesla T4 and Tesla P4 (TensorRT), Atlas 300 (ACL), Snapdragon 845 Hexagon DSP (SNPE), Raspberry 3B (TVM) and Intel i7-6700 (FBGEMM). We

**Table 8:** Accuracy comparison of LSQ on MobileNetV2, given different $L2$ regularization of the scale parameter.

| Weight Decay | $1e-5$ | $2e-5$ | $4e-5$ | $6e-5$ | $8e-5$ | $1e-4$ | $2e-4$ |
|---|---|---|---|---|---|---|---|
| Accuracy | 66.10 | 67.04 | 67.86 | 68.30 | **68.41** | 68.03 | 66.80 |

**Table 9: Latency Benchmark** with different architectures and different hardware under 8 bits. All results are testd with 5 runs. (Numbers in the bracket represent the speed up ratio compared with the former row. Green means faster and Red means slower.)

| | Model | | ResNet-18 | ResNet-50 | MobileNetV2 | EfficientNet-Lite0 | RegNetX-600MF |
|---|---|---|---|---|---|---|---|
| | FLOPS | | 1813M | 4087M | 299M | 385M | 600M |
| **Batch 1** **Latency** **(ms)** | Tesla T4 | FP32 | 1.27 | 3.33 | 0.94 | 1.39 | 1.69 |
| | | FP16 | 0.54(2.35) | 1.11(3.0) | 0.42(2.24) | 0.49(2.84) | 1.24(1.36) |
| | | INT8 | 0.43(1.26) | 0.89(1.25) | 0.43(0.98) | 0.49(1.0) | 1.39(0.89) |
| | Tesla P4 | FP32 | 1.98 | 4.67 | 1.82 | 2.19 | 4.33 |
| | | INT8 | 0.99(2.0) | 2.08(2.25) | 1.03(1.77) | 1.53(1.43) | 3.7(1.17) |
| | Atlas300 | FP16 | 1.25 | 2.52 | 103.2 | 97.31 | 1.54 |
| | | INT8 | 1.0(1.25) | 1.96(1.29) | 102.2(1.01) | 95.92(1.01) | 1.39(1.11) |
| | Snapdragon 845 | FP32 | 15.13 | 36.04 | 9.27 | 11.09 | 20.84 |
| | | INT8 | 12.09(1.25) | 25.64(1.41) | 9.02(1.03) | 10.51(1.06) | 16.69(1.25) |
| | Raspberry 3B | FP32 | 689.50 | - | 200.30 | - | 252.09 |
| | | INT8 | 567.33(1.22) | - | 168.73(1.19) | - | 230.77(1.09) |
| | Intel i7-6700 | FP32 | 30.75 | 86.63 | 184.85 | 238.33 | 22.39 |
| | | INT8 | 16.42(1.87) | 20.22(4.28) | 9.16(20.18) | 10.79(22.09) | 9.06(2.47) |
| **Batch 8** **Latency** **(ms)** | Tesla T4 | FP32 | 6.21 | 15.85 | 3.29 | 4.59 | 4.34 |
| | | FP16 | 1.68(3.7) | 4.17(3.8) | 1.35(2.44) | 2.26(2.03) | 2.59(1.68) |
| | | INT8 | 0.79(2.13) | 2.28(1.83) | 0.88(1.53) | 1.03(2.19) | 2.43(1.07) |
| | Tesla P4 | FP32 | 6.69 | 18.42 | 5.54 | 7.21 | 6.52 |
| | | INT8 | 3.02(2.22) | 6.25(2.95) | 2.24(2.47) | 4.69(1.54) | 4.59(1.42) |
| | Atlas300 | FP16 | 5.43 | 13.59 | 745.34 | 750.73 | 4.47 |
| | | INT8 | 4.37(1.24) | 10.82(1.26) | 850.91(0.88) | 667.41(1.12) | 5.15(0.87) |
| | Snapdragon 845 | FP32 | 143.55 | 326.19 | 46.56 | 54.89 | 121.80 |
| | | INT8 | 77.18(1.86) | 168.28(1.94) | 37.97(1.23) | 44.6(1.23) | 76.67(1.59) |
| | Intel i7-6700 | FP32 | 137.94 | 350.68 | 409.48 | 522.09 | 75.79 |
| | | INT8 | 58.33(2.36) | 124.14(2.82) | 33.0(12.41) | 44.25(11.8) | 38.35(1.98) |

choose five prevalent network architectures: ResNet-18, ResNet50, MobileNet-V2, Efficient-Lite0, RegNetX-600MF. Batch size is set as 1 and 8. Among the diverse hardware, Raspberry 3B is an edge device with limited computational resource and requires a long compilation process with TVM for 8-bit. So we only test ResNet-18, MobileNet-V2 and RegNetX-600MF with batch size of 1.

The overall results are listed in Table 9. The number in bracket displays the speed up compared with the former row (Green represents faster and Red means slower). It can be seen that most cases can enjoy a satisfactory acceleration. However, there still exists cases with little gains and some are even slower with quantization. This indicates that some hardware drops behind for novel network architectures. For GPU such as Tesla T4 and P4, there is a consistent improvement with INT8 compared with FP32, especially for the ResNet-family. However, because Tesla T4 introduces Tensor Core supporting FP16, INT8 shows little advantage compared with FP16. Atlas of HUAWEI is the representative of ASIC. Its INT8 optimization seems very preliminary and only have a 1.03%~1.29% speed up. For the searched advanced model EfficientNet-Lite0 and RegNetX-600MF, it is even slower using 8-bit for batch 8. For the Mobile DSP on Snapdragon 845, we find that it can ensure a more efficient inference with 8-bit no matter what network we choose. For batch 8, the speed up ratio is up to around 2. Beside, we also evaluate CPU including X86 and ARM. For X86 CPU, we directly utilize the official implementation from Facebook and the FBGEMM's INT8 inference can save up to 20 times of latency compared with the default FP32 counterpart of PyTorch. For ARM CPU, we only compile the integer convolutional kernel with 200 iterations and they can already achieve a 9%~22% improvement.

With the comprehensive evaluation of latency, researchers can acquire the knowledge of practical acceleration brought by quantization instead of remain the level of theory. Meanwhile, hardware vendors can identify the under-optimized network architecture and put in more efforts. We believe both communities will benefit from it.

# C Post-Training Quantization.

## C.1 Definition of different calibration algorithm

Calibration is the offline process to calculate scale and zero points of activations and weights by simple forward of pretrained floating-point models on a sampled dataset. Usually, different calibration algorithm employs different statistics like maximum or mean for the calculation. No training is involved in this process which is simple and practical. With $t$ bit-width, the floating-point $\boldsymbol{x}$ is quantized into $\bar{\boldsymbol{x}}$, the definition of calibration algorithm to calculate scale $s$ is as follows:

**MinMax calibration.** The minimum value and maximum value of the original floating-point $\boldsymbol{x}$ are recorded to calculate scale and zero points. For symmetric quantizer, the equation is as follows:

$$s = \frac{max(|\boldsymbol{x}|)}{(2^t - 1)/2} \tag{6}$$

**Quantile calibration.** The quantile $\alpha$ is used to determine the maximum value and minimum value. Unless specified noted, the quantile is set to 0.9999 in default. Quantile is a hyper-parameter which means $(1 - \alpha)\%$ largest values are clipped. In asymmetric quantizer, the equation is as follows:

$$s = \frac{quantile(\boldsymbol{x}, \alpha) - quantile(\boldsymbol{x}, 1 - \alpha)}{2^t - 1} \tag{7}$$

**MSE calibration.** The maximum value is searched to minimize mean squared quantization to find a better clip range for the calculation of scale and zero point. For symmetric or asymmetric quantizer, the equation is as follows:

$$\min_s \|\boldsymbol{x} - \bar{\boldsymbol{x}}\|^2 \tag{8}$$

**KLDivergence calibration.** The maximum value is also searched to minimize the KL divergence of two distribution between the original FP $\boldsymbol{x}$ and the quantized $\bar{\boldsymbol{x}}$. To get the distribution of $\boldsymbol{x}$ and $\bar{\boldsymbol{x}}$, we use the histogram of data and split it into 2048 bins denoted as $hist(*)$. For symmetric or asymmetric quantizer, the equation is as follows:

$$\min_s D_{kl}(hist(\boldsymbol{x}), hist(\bar{\boldsymbol{x}})) \tag{9}$$

**Norm calibration.** The p-norm of absolute value $|x|$ scaled by the maximum quantization number is $N_{max}$ is used to compute scale. Norm calibration is introduced in LSQ [15] with $p = 2$. Unless specified noted, l2 norm is used in default. For symmetric quantizer, the equation is as follows:

$$s = \frac{\||x|\|^p}{\sqrt[2]{N_{max}}} \tag{10}$$

**MeanStd calibration.** The mean and standard deviation denoted as $\mu$ and $\sigma$ of the original floating point $\boldsymbol{x}$ are recorded to calculate quantization scale. $\alpha$ is a hyper-parameter, LSQ+ [42] uses 3 and DSQ [28] uses 2.6. Unless specified noted, 3 is used in default. For symmetric quantizer, the equation is as follows:

$$s = \frac{max(|\mu - \alpha \times \sigma|, |\mu + \alpha \times \sigma|)}{(2^t - 1)/2} \tag{11}$$

## C.2 PTQ experiments

We conduct extensive experiments on different calibration algorithms. For brevity, only MinMax, MSE, KLD, and Quantile are reported in Table 10. 5 models are evaluated to compare the quantization sensitivity among different architectures. Among the results presented below, we find no calibration algorithm always works the best. However, as the calibration requires very little time cost, evaluation with several different calibration algorithms helps us to find some useful insights for algorithm and hardware designers.

**Efficient models tend to be more sensitive to quantization.** While ResNet-18 and ResNet-50 are almost lossless with simple calibration in their best setting. MobileNetV2 and EfficientNet-Lite0 have 1.4% and 1.6% accuracy degradation even with their best setting. However, RegNetX-600MF is also lossless. We assume that group convolution in RegNetX-600M is more robust to quantization than depthwise convolution in MobileNetV2 and EfficientNet-Lite0.

**Different networks may favor different calibration algorithms.** In inference library SNPE, MinMax for both activation and weights is the best among four different settings for ResNet-18, ResNet-50, and RegNetX-600MF.

**Table 10: PTQ** Post-training quantization benchmark on the ImageNet dataset with different calibration algorithm. The accuracy is reported with the mean and variance of accuracy after 3 runs. Two inference library named SNPE and FBGEMM are reported. SNPE has per-tensor quantizer for weights while FBGEMM has per-channel quantizer.

| Model | Bit | A_calibration | W_calibration | SNPE Acc. | FBGEMM Acc. |
|---|---|---|---|---|---|
| ResNet-18 FP:71.0 | 8 | MinMax | MinMax | **70.7 ± 0.05** | **70.9 ± 0.01** |
| | 8 | MSE | MinMax | **70.7 ± 0.04** | 70.8 ± 0.04 |
| | 8 | KLD | MinMax | 69.5 ± 0.01 | 69.7 ± 0.05 |
| | 8 | Quantile | MinMax | 66.6 ± 0.16 | 66.8 ± 0.2 |
| ResNet-50 FP:77.0 | 8 | MinMax | MinMax | **76.5 ± 0.02** | **76.6 ± 0.03** |
| | 8 | MSE | MinMax | 76.4 ± 0.001 | **76.6 ± 0.03** |
| | 8 | KLD | MinMax | 75.7 ± 0.02 | 75.9 ± 0.04 |
| | 8 | Quantile | MinMax | 75.8 ± 0.06 | 76.0 ± 0.04 |
| MobileNetV2 FP: 72.6 | 8 | MSE | MSE | **71.1 ± 0.2** | **71.2 ± 0.04** |
| | 8 | MinMax | MinMax | 70.4 ± 0.04 | 70.9 ± 0.1 |
| | 8 | MSE | MinMax | 70.9 ± 0.09 | **71.2 ± 0.06** |
| | 8 | KLD | MinMax | 69.3 ± 0.04 | 70.0 ± 0.15 |
| | 8 | Quantile | MinMax | 70.0 ± 0.03 | 70.6 ± 0.1 |
| EfficientNet-Lite0 FP: 75.3 | 8 | MSE | MSE | **72.6 ± 0.03** | **73.7 ± 0.06** |
| | 8 | MinMax | MinMax | 71.0 ± 0.05 | 73.4 ± 0.01 |
| | 8 | MSE | MinMax | 71.5 ± 0.09 | **73.7 ± 0.05** |
| | 8 | KLD | MinMax | 48.1 ± 0.82 | 72.2 ± 0.02 |
| | 8 | Quantile | MinMax | 67.0 ± 0.9 | 71.4 ± 0.2 |
| RegNetX-600MF FP: 73.7 | 8 | MSE | MSE | 73.3 ± 0.02 | 73.4 ± 0.01 |
| | 8 | MinMax | MinMax | **73.4 ± 0.05** | **73.5 ± 0.04** |
| | 8 | MSE | MinMax | 73.3 ± 0.04 | **73.5 ± 0.03** |
| | 8 | KLD | MinMax | 72.1 ± 0.02 | 72.2 ± 0.02 |
| | 8 | Quantile | MinMax | 71.4 ± 0.2 | 71.4 ± 0.2 |

**Table 11: Initialization** PTQ accuracy is reported with simple calibration algorithm. The accuracy after BatchNorm calibration is denoted as BN Calib Acc. QAT Acc is reported with the same quantization-aware training settings.

| Model | Bit | Setting | A_calibration | W_calibration | PTQ Acc. | BN Calib Acc. | QAT Acc. |
|---|---|---|---|---|---|---|---|
| MobileNetV2 FP: 72.6 | 4 | Academic | Quantile | Norm | 1.1 | 51.3 | 70.7 |
| | 4 | Academic | Quantile | MeanStd | 0.76 | 46.1 | 70.7 |
| | 4 | Academic | Norm | Norm | 0.88 | 45.0 | 70.4 |
| | 4 | Academic | MSE | MeanStd | 0.65 | 39.8 | 70.3 |
| | 4 | Academic | MinMax | MinMax | 0.43 | 28.1 | 69.0 |
| | 4 | SNPE | Quantile | Norm | 0.13 | 0.14 | 65.2 |
| | 4 | SNPE | Norm | Norm | 0.18 | 0.18 | 67.0 |

In efficient models, using MSE for both activations and weights is the best MobileNetV2 and EfficientNet-Lite0. MSE also works fines for RegNetX-600MF. It indicates that compared with using the maximum value, searching for a better clipping value is essentially important for an efficient model which may contain some outliers.

**Different calibration algorithms bring a large accuracy gap.** For ResNet-18, MSE calibration for activation is better than Quantile with over 4.1% with the same MinMax calibration for weights. Similarly for EfficientNet-Lite0, using MSE calibration for weights is better than using MinMax with over 1.1% accuracy gain with the same MSE calibration for activation.

**Per-channel quantizer tends to have higher accuracy up bounds.** We also compare a per-channel quantizer in FBGEMM with a per-tensor quantizer in SNPE. Although the advantage is not obvious in heavy models, the accuracy of per-channel is higher with 1.1% than per-channel for EfficientNet-Lite0. The results reveal that a per-channel quantizer for weights may come as a better design for increasing the PTQ accuracy.

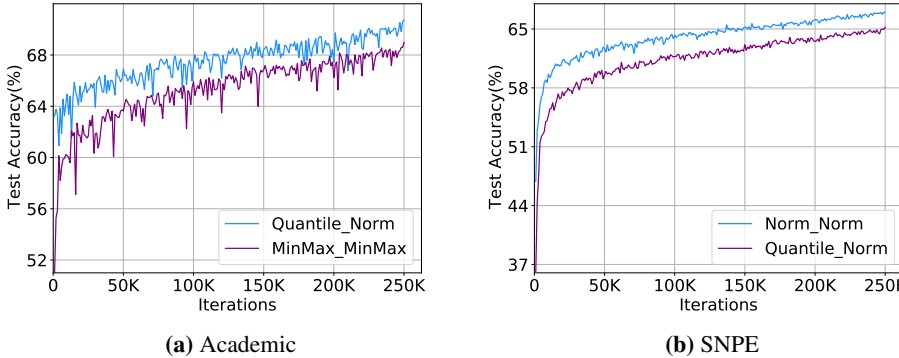

**Figure 14:** Comparison of different initialization. The accuracy curve is plotted and shows that better initialization results in fast convergence and better final accuracy. Quantile_Norm denotes Quantile calibration for activation and Norm calibration for weights.

## C.3 The impact of BatchNorm calibration.

In the PTQ experiments under 4 bit, we find that simple calibration fails with accuracy under 1%. The results are listed below in Table 11. Therefore, as suggested by recent papers like [49], BatchNorm statistics changes after quantization especially for low-bit models. Therefore, we propose a simple BatchNorm statistics calibration to see the impact of BatchNorm calibration.

Similar to the calibration of quantization parameters, We sample a calibration dataset to perform the calibration of BatchNorm statistics. The calibration dataset is composed of 4096 training images which are randomly sampled. In the initialization of quantization parameters, the calibration dataset is forward to calculate scale and zero points, which follows the definition of the different calibration algorithms. And then the quantization parameters are kept unchanged, we forward the calibration dataset again to recalculate the BatchNorm statistics as the method proposed in [50]. The running mean and variance of BatchNorm are reset with the average mean and variance in the calibration dataset. In a 4bit academic setting, BatchNorm statistics calibration brings large accuracy improvement. With Quantile calibration for activation and Norm calibration for weights, the accuracy is 51.3% after simple BatchNorm calibration. The Norm calibration for both activation and weights is the same as the LSQ [15] proposes with 45.0% after BN. Therefore, we find a better initialization method for LSQ with over 6.3% accuracy improvement. However, BatchNorm calibration fails under hardware settings. Th BatchNorm Calibration accuracy is still below 1%. It indicates that a more advanced PTQ algorithm is needed for this case.

## C.4 The impact of initialization for QAT

While using different calibration algorithms is a common practice for post-training quantization to improve accuracy, it is rarely discussed in quantization aware training. The recent approach named LSQ+ [42] proposes a different calibration named MeanStd for weights and MSE for activations to initialize scale and offset, it shows that different initialization of quantization parameters affects the final QAT accuracy on EfficientNetB0. We extend their experiments with more calibration algorithms with different inference libraries as shown in Table 11.

With the same training set and different calibration algorithms for initialization, the initialization accuracy varies with large gaps. In 4 bit settings, the initialization accuracy using Norm is better than MinMax by 16.9%, and after a full training pipeline with 100 epochs, the QAT accuracy of Norm still surpasses MinMax by 1.4%. However, the final accuracy of Quantile for activation and Norm for weights is only slightly better with 0.3% than Norm for activation and Norm for weights. In hardware settings, the initialization accuracy of different calibration algorithms is below 1%. Therefore, we choose the top1 setting and the original LSQ initialization from the academic setting to perform this comparison. The Quantile is lower than Norm initialization by 1.8%. The training curve is shown in Fig. 14, the right calibration algorithm is chosen to initialize the quantization parameters, it can bring accuracy improvements and bring fast convergence. The calibration algorithm acts as the bridge between PTQ and QAT. Our experiments verify that the combination of the PTQ and QAT process is crucial and affects the final QAT accuracy. How to combine PTQ and QAT still needs further discussion.

# D Related Work

**Quantization Algorithms.** As an appealing strategy for model compression and acceleration, quantization algorithm has driven much attention ever since 90s in last century. In (Holi & Hwang, 1993) [51], an empirical

analysis on simple neural networks show 8-bit is sufficient for lossless quantization, Hoehfeld & Fahlman (1992) [52] developed a stochastic rounding scheme to further quantize artificial neural network below 8-bits. In modern deep learning, low-bit quantization has driven lots of interests in academical research. DoReFa-Net [31] first proposes to quantize neural networks into multi-bit. [15, 18, 30] leverage the straight-through estimator (STE) to learn an appropriate clipping range for quantization. DSQ [28] studies the variant of STE and approximate the gradient of step functions. Works like [29, 53, 54] use non-uniform quantization to better adapt the distribution of the weights and activations. Recently, mixed-precision algorithm also made significant progress. For example, [44] uses reinforcement learning to study the bit-width allocation. The eigenvalue of Hessian matrix is employed in [55] to determine the bit for each layer. Differentiable methods like [56, 57, 58, 59] directly learns the bit-width through gradient descent.

Along with quantization-aware training (QAT), a much easier way to perform quantization is called post-training quantization (PTQ). PTQ only requires 10-1000 training images and tiny computational resources to finish the quantization. [60, 61, 62] seek to find the optimal clipping range for 4-bit PTQ. OCS [63] uses channel-splitting to deal with outliers. [5, 45] even do quantization without accessing any real data. [64, 65] adopt intermediate feature-map reconstruction to optimize the rounding policy.

**Quantization Frameworks and Hardware.** Generally speaking, hardware for quantization can be naturally divided to specialized hardware and general hardware. In 1990, Hammerstrom [66] already designed specialized VLSI hardware for 8-bit and 16-bit training. More recently, specialized hardware like BISMO [67] and BitFusion [68] are designed to handle mixed-precision inference. On general hardware, NVIDIA Volta Tensor Cores [69] can support FP16 & FP32 mixed training and achieve at least 2x speed-up. As for low-bit inference, there are several hardware libraries: such as NVIDIA's TensorRT [22], Qualcomm's SNPE [24], and so on. Hardware providers also build some framework for model quantization (or other compression techniques). For example, the Neural Network Compression Framework (NNCF) [70] developed by Intel supports quantization and pruning. However, their implementation can only be shared on OpenVINO. AIMET, developed by Qualcomm, also just focuses on their own hardware. To our best knowledge, no existing framework can handle multi-platform deployment of quantization neural networks with diverse advanced algorithms.

**Benchmarks.** In many sub-fields of deep learning or computer vision, a thorough analysis and evaluation is necessary [71]. In neural architecture search (NAS) [72], the search space is notoriously large and the search process requires tremendous computation resources, making researchers hard to reproduce results of prior works. For this reason, NAS-Bench-101 [19] and -201 [48] are proposed to solve the reproducibility issues and make a fair evaluation for NAS research. Recently, Hardware-NAS-Bench [73] was proposed to benchmark the inference latency and energy on several hardware devices. Despite the progress of benchmarks in NAS and other areas, model quantization lacks such standard to foster the reproducibility and deployability.

# E   Detailed Inference Library Setup

**Academic Setup.** In academical research, most existing work chooses the per-tensor, symmetric quantization. This quantizer design could be challenging. However, academical setting only quantizes the input and the weight of a convolutional or linear layer. Thus the computational graph is not aligned to any hardware implementations. Note that people tend to use *unsigned* quantization to quantize input activation and *signed* quantization to quantize weights. For unsigned quantization, the target integer range is $[N_{min}, N_{max}] = [0, 2^t - 1]$, while for signed quantization, the range becomes $[N_{min}, N_{max}] = [-2^{t-1}, 2^{t-1} - 1]$. The intuition for adopting unsigned quantization is ReLU activation are non-negative, and symmetric signed quantization will waste one bit for negative parts. In our implementation, we add a switch variable called *adaptive signness*, which can turn the integer range to $[-2^{t-1}, 2^{t-1} - 1]$ based on data statistics. It should be noted that *Adaptive signness* is only designed for academic setting, while symmetric quantization must waste one bit for non-negative activation in real-world hardware.

**TensorRT Setup.** TensorRT [22] is a high-performance inference library developed by NVIDIA. The quantization scheme in TensorRT is symmetric per-channel for weights, and symmetric per-tensor for activations. The integer range is $[-128, 127]$. TensorRT will quantize every layers in the network including Add and Pooling besides those layers which have weights. However, per-channel quantization scheme can reduce the error for weight quantization to a certain extent. TensorRT model will be deployed on GPUs, and Int8 computation will be achieved by Tensor Cores or DP4A instructions, which are highly efficient. Typically, one GTX1080TI GPU have 45.2 peak Tops in INT8 mode.

**SNPE Setup.** SNPE is a neural processing engine SDK developed by Qualcomm Snapdragon for model inference on Snapdragon CPU, Adreno GPU and the Hexagon DSP. SNPE supports 8bit fixed-point quantization for Hexagon DSP. Hexagon DSP is an advanced, variable instruction lenghth, Very Long Instruction Word(VLIW) processor architecture with hardware multi-threading. The quantization scheme of SPNE is asymmetric and per-tensor for weights and activations.

**TVM Setup.** TVM [25] is a deep learning compiler and can compile neural networks to a 8-bit implementation. For now, TVM supports running the whole network in symmetric per-tensor quantization scheme. One different point is, in order to accelerate the quantization affine operation, they represent the scale as power-of-two and thus can utilize the efficient shift operation to enjoy further speed up. The quantized INT8 model compiled by TVM can be deployed on GPUs, CPUs or DSPs.

**ACL Setup.** ACL is a neural network inference software developed by HUAWEI for the hardware named Atlas. Atlas supports INT8 convolution and linear kernel so ACL quantizes the layer which has weights, such as convolution, fully-connected layer to int8 fixed point, but remains the rest part of network in FP32. The quantization scheme is symmetric per-channel for weight, and asymmetric per-tensor for activation to avoid the waste of one bit. Typically an Atlas 300 inference card have 88 Tops in INT8 mode.

**FBGEMM Setup.** FBGEMM is a inference library developed by Facebook and can deploy torch model easily. The quantization scheme is asymmetric per-channel and we quantize the whole network into int8 fixed point.

# F    Quantization Algorithms Implementation

**Learned Step Size Quantization.** LSQ leverages the Straight-Through Estimator [74] to learn the quantization scale for each layer. For initialization, we use the method proposed in original paper: the scale is determined by $s = 2||\boldsymbol{w}||_1/\sqrt{N_{max}}$. For symmetric quantization, the zero point is initialized to 0, and kept fixed. For asymmetric quantization, zero point is initialized to $N_{min}$ if the activation is non-negative. Inspired by LSQ+ [42], the zero point can also be updated through backpropagation with the help of STE. Therefore we make it learnable in asymmetric quantization. LSQ uses gradient scale to stabilize the scale learning. The gradient scale is determined by $1/\sqrt{MN_{max}}$ where $M$ is the number of elements in that tensor. We extend this gradient scale to per-channel weight learning, where the $M$ is the number of weights in each filter.

**Differentiable Soft Quantization.** DSQ uses the hyperbolic tangent function to approximate the conventionally adopted STE. In our implementation, we use $\alpha = 0.4$ (for definition please refer to the original paper [28]) which controls the shape and smoothness of the $\tanh$ function. For weight quantization, we use the min-max range as

$$Clip_{min} = \mu(\boldsymbol{w}) - 2.6\sigma(\boldsymbol{w}), \tag{12}$$

$$Clip_{max} = \mu(\boldsymbol{w}) + 2.6\sigma(\boldsymbol{w}), \tag{13}$$

where $\mu(\cdot)$ and $\sigma(\cdot)$ computes the mean and standard deviation of the tensor. Then, the scale is determined by $s = \frac{\max(-Clip_{min}, Clip_{max})}{N_{max}-N_{min}}$ for symmetric quantization, and $s = \frac{Clip_{max}-Clip_{min}}{N_{max}-N_{min}}$ for asymmetric quantization. The zero point is set to 0 for symmetric and $N_{min} - \lfloor \frac{Clip_{min}}{s} \rceil$ for asymmetric quantization. For activation, we use the BatchMinMax as the clipping range, i.e. the averaged min-max range across the batch dimension. This is further updated with exponential moving average across different batches with momentum 0.9, similar to BN.

**Parameterized Clipping Activation.** PACT is introduced to quantized activation by learning the clipping threshold through STE. Its activation is clipped by a parameter $\alpha$ first. Then, the clipped activation is quantized and re-quantized. Although PACT and LSQ both learns the scale, they have three differences. First, the clipping range in PACT is handcrafted initialized to 6 while LSQ initialization is based on the tensor $L1$ norm. Second, PACT has no gradient in the range of clipping. While LSQ can compute the gradient. Third, PACT does not scale the gradient of $\alpha$, while LSQ does. Note that PACT only has non-negative, unsigned quantization in the first. To extend it to our hardware settings, we clip the activation to $(-\alpha, \alpha)$ in symmetric case and $(\beta, \alpha)$ for asymmetric case, (where $\beta$ is initialized to -6). For weight quantization of PACT, it is the same with DoReFa-Net.

**DoReFa-Network.** DoReFa-Net simply clips the activation to $[0, 1]$ and then quantizes it. This is based on the intuition that most activation will fall into this range in old network architectures, e.g. AlexNet [75] and ResNet [17]. In hardware settings, we modify the activation range to $[-1, 1]$ for both symmetric and asymmetric quantization. As for weight quantization, it can be described as:

$$\tilde{\boldsymbol{w}} = \tanh(\boldsymbol{w})\frac{1}{\max(|\tanh(\boldsymbol{w})|)}, \tag{14}$$

$$\hat{\boldsymbol{w}} = \text{dequantize}(\text{quantize}(\tilde{w})), \tag{15}$$

where the first step is a non-linear transformation and the second step is the same with Eq. (1). The scale is simply calculated by $\frac{2}{N_{max}-N_{min}}$ for symmetric quantization and $\frac{\max(\tilde{\boldsymbol{w}})-\min(\tilde{\boldsymbol{w}})}{N_{max}-N_{min}}$ for asymmetric quantization.

**Additive Powers-of-Two Quantization.** APoT quantization uses multiple PoT's (Powers-of-Two) combination to composes a set of non-uniform quantization levels. Since the quantization are non-uniform in most cases (except the case of 2-bit the APoT becomes uniform quantization), we do not benchmark it on real hardware. Additionally, APoT introduces weight normalization (similar to standardization [76] technique) to smooth the learning process of clipping range in weight. However, it is unclear how to incoporate this technique with

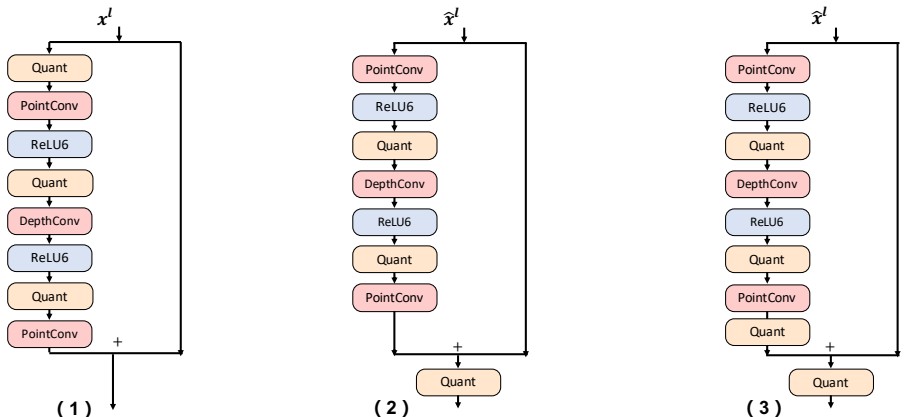

**Figure 16:** Comparison of different quantization implementations for inverted bottleneck block in MobileNetV2 [13].

BN folding. Therefore, we only reproduce it in our academic setting. The implementation are based on the open-source codes.

**Quantization Interval Learning.** QIL composes of two unit to quantization: (1) the first one is called transformer, which transform the weights or activation to $[-1, 1]$ ($[0, 1]$ as for non-negative activation). This transformer also has two functionalities: pruning and non-linearity. (2) The second one is called quantizer, given by

$$\tilde{\boldsymbol{w}} = \text{clip}\left((\alpha|\boldsymbol{w}| + \beta)^\gamma, 0, 1\right) * \text{sign}(\boldsymbol{w}), \tag{16}$$

$$\hat{\boldsymbol{w}} = \text{dequantize}(\text{quantize}(\tilde{\text{w}})), \tag{17}$$

where $\alpha = \frac{1}{2*D}$ and $\beta = -\frac{C}{2D} + \frac{1}{2}$. This transformation maps the weight from $[C - D, C + D]$ to $[0, 1]$ and $[-C - D, -C + D]$ to $[-1, 0]$. As a result, the weights between $[-C + D, C - D]$ are pruned. The non-linearity of the transformation function is introduced by $\gamma$. This parameter can control the linearity and thus control the quantization interval. However, we find this technique is extremely unstable. In our experimental reproduction, learning $\gamma$ will not converge. In the original paper, the gradient scale of $C$ and $D$ is set to 0.01. We find this gradient scale also leads to frequent crashes. Thus we use the gradient scale introduced in LSQ, i.e. $\frac{1}{\sqrt{MN_{max}}}$.

# G  Block Graph

In this section we provide visualization of the graph implementation. First, we visualize concatenation operation. In academic setting, the concatenation is not considered to be quantized and is operated at FP32. However, in real-world hardware we must quantize the input of the concatenation. See Fig. 15 aside. It is worthwhile to note that there is only one implementation for concatenation in all hardware deployment environments. Moreover, the quantization of two branches must share one set of scale and zero point. Other-wise the fused branch will be represented with higher precision. This parameter-sharing mechanism in concatenation may cause severe accuracy degradation in practice, although we do not testify any architectures containing concatenation operations.

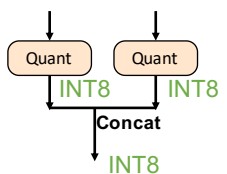

**Figure 15:** Concatenation graph.

Then, we visualize the *inverted bottleneck block* implementation adopted in MobileNetV2 and EfficientNet-Lite0. As shown in Fig. 16, graph 1 stands for academic setting where only the input of a convolutional layer is quantized. Graph 2, 3 describes the graph implementation in real-world hardware. The difference is similar to Basic Block in Fig. 4, where graph 2 omits the quantization of the main branch in shortcut-add.

