# OpenReview forum: "MQBench: Towards Reproducible and Deployable Model Quantization Benchmark"
_NeurIPS.cc/2021/Track/Datasets_and_Benchmarks/Round1 — NeurIPS 2021 Datasets and Benchmarks Track (Round 1)_

### Official Review · Reviewer_dTK1 · 2021-06-28
**This paper introduces a novel benchmark to evaluate quantized models. Its main goal is to benchmark quantization methods using reproducible setups i.e. with the identical hyperparameters, models, tasks, hardware.**

**Rating:** 6
**Confidence:** 4

**Strengths:**

1. Reproducibile research with fair evaluations is important for the community.

2. MQBench methodology is clearly described. Also, paper discusses corner cases during training and inference phases of quantized models.

3. Evaluation of existing algorithms show interesting results that can be valuable for broader research community.

**Weaknesses:**

1. The biggest weakness of this paper is lack of documented code and a web-site with public submission form. Looks like some code has been released, but there is no documentation and step-by-step guide how to reproduce MQBench results. Also, it is not possible for the community to submit their QAT methods for fair comparisons. Therefore, the main goal of this paper is not fulfilled unless authors release clearly described documentation how to run their code and/or submission web-site is provided.

2. Currently, only ImageNet classification has been used in MQBench. It means that MQBench is limited to a particular task and a dataset. Authors recognize this limitation and suggest object detection and NLP applications as a potential future work.

3. I believe, comparison to MLPerf/MLCommons benchmarks is missing in this paper.

**Additional Feedback:**

I suggest authors to release documentation for their code with step-by-step guide on how to reproduce each experiment, and finish their web-site with public submission form. Then, I will increase paper score.

Upd 7/13: I increased my score based on author's feedback. They made a good effort to improve benchmark web-site, documentation and code reproducibility.

**Clarity:**

This paper is well written. However, some sentences need minor revisions:
#130: a activation -> an activation
Table 5: asynchronized -> asynchronous


**Correctness:**

Methodology is correct, but has some limitations: error bars are missing, only a single classification task on a single ImageNet dataset has been evaluated.

**Documentation:**

No, benchmark is not reproducible in a current form

**Relation To Prior Work:**

This paper clearly discusses differences with previous contributions. Though some discussion on other benchmarks such as MLPerf/MLCommons, I believe, is missing.

**Summary And Contributions:**

Authors propose a novel benchmark called MQBench to analyze and evaluate quantized by quantization-aware training (QAT) models using consistent setups. Their motivation is to fairly compare existing and future QAT methods with the same hyperparameters. Another goal of this paper is to benchmark QAT models with real-world deployments on CPU, GPU, ASIC, DSP (using corresponding vendor libraries). Authors clearly describe the methodology for benchmarking, batch normalization folding, and quantized convolutional layer implementations. Unfortunately, this paper falls short of fulfilling reproducible research goals due to unfinished code documentation and benchmark web-site.

---

> ### Author Response · Authors · 2021-07-13
> **Response to Reviewer dTK1**
>
> Thank you for your thoughtful review. Please see our detailed response below
>
> Q1: *The biggest weakness of this paper is lack of documented code and a web-site with public submission form. Looks like some code has been released, but there is no documentation and step-by-step guide how to reproduce MQBench results. Also, it is not possible for the community to submit their QAT methods for fair comparisons. Therefore, the main goal of this paper is not fulfilled unless authors release clearly described documentation how to run their code and/or submission web-site is provided.*
>
> A1: Thank you for your advice, we have released our documentation, website, and GitHub repository, which contains a step-by-step guide to reproduce MQBench and add new algorithms. For researchers who intend to submit their results, they can submit a merge request that contains the implementation, log file, running scripts to our repo.
>
> Q2: *Currently, only ImageNet classification has been used in MQBench. It means that MQBench is limited to a particular task and a dataset. Authors recognize this limitation and suggest object detection and NLP applications as potential future work.*
>
> A2: Yes, that could be a limitation of MQBench. We do not run experiments on detection mainly because there exist few baseline results on these tasks. To be honest, we are implementing MQBench on MMDetction right now but it is still under development. We will keep on implementing more algorithms, more tasks, more datasets, and also welcome community contribution.
>
> Q3: *I believe, comparison to MLPerf/MLCommons benchmarks is missing in this paper*
>
> A3: Good point, comparison to MLPerf is needed. The main difference between our MQBench and MLPerf is the evaluation metric. MLPerf/MLCommons are designed to evaluate the training/inference efficiency on a  **system of software (ML framework) and hardware**. It does not compare the accuracy as long as the model reaches a certain criterion. However, MQBench is designed to evaluate the **accuracy of algorithms under different hardware**, we do not compare the hardware performance.

---

### Official Review · Reviewer_aXe9 · 2021-06-30
**MQBench**

**Rating:** 6
**Confidence:** 3

**Strengths:**

1. This work evaluates the reproducibility of several methods in a single benchmark tool. Reproducibility is a significant challenge and direct algorithm comparison in a single tool can help advance the state of the art within the community.
2. The introduction presents strong arguments and outlines the need for a benchmarking tool within the community.
3. The paper is well organized with clear tables and diagrams.
4. Clear results and recommendations are made after the data is presented.
5. Detailed evaluation of how BN folding impacts accuracy vs. performance showed interesting results.

**Weaknesses:**

1. Authors claim the code is open source, but the link provided did not have any code posted. Please include a link to the open-source software or remove the open-source check from Table 1.
2. Very few details were provided on how the framework functions (such as input/output/options/system requirements/etc.)
3. How extensible is the framework to new hardware and/or new deep learning layers?

**Additional Feedback:**

Minor grammar errors throughout such as lines 41, 44, 110.

**Clarity:**

The paper is organized in a clear and straightforward way. The tables and diagrams are well-organized and concise.

**Correctness:**

Results in the supplementary material provide strong evidence to support the finding presented within the paper. I have no reason to believe claims are not accurate.

**Documentation:**

The appendix contains a detailed list of hyperparameters needed to reproduce the assessments. As mentioned earlier, frameworks would benefit from a brief discussion on the capabilities and limitations.

**Ethics:**

No concerns identified

**Relation To Prior Work:**

This paper benchmarks several methods from prior work. The goal of this research is to reproduce prior work as well as assess each algorithm's ability to deploy in standard hardware. All prior work is referenced or sourced.

**Summary And Contributions:**

This work attempts to reproduce recent quantization research via a benchmarking tool called MQBench. The authors test a variety of algorithms and assess the real-world deployability of each method. MQBench connects algorithms to hardware to assess the performance and accuracy gained or lost through quantization.

---

> ### Author Response · Authors · 2021-07-13
> **Response to Reviewer aXe9**
>
> Thank you for your thoughtful review. Please see our detailed response below
>
> Q1: *Authors claim the code is open source, but the link provided did not have any code posted. Please include a link to the open-source software or remove the open-source check from Table 1.*
>
> A1: Please see our general response, we have released our documentation, GitHub repository, and website.
>
> Q2: *Very few details were provided on how the framework functions (such as input/output/options/system requirements/etc.)*
>
> A2: Sorry for the lack of details, in our revised submission, we add a new section describing the implementation of MQBench. You can also find it in our GitHub repo, where we provide all running scripts in the submission and the systems requirements for installing MQBench.
>
> Q3: *How extensible is the framework to new hardware and/or new deep learning layers?*
>
> A3: Good question. For new hardware, it is easy to extend if it only changes the quantizer setting. For example, in our GitHub README, we give an example to extend to TFLite Micro. For new deep learning layers, you just need to add the corresponding fake quantization function if the hardware library has inference support. If not, the users have to define a real quantization routine besides fake quantization to guide the hardware design. We will release a guide to demonstrate how to add a new layer.

---

### Official Review · Reviewer_iQFC · 2021-07-06
**Good overview of NN quantization, but lacking as a benchmark**

**Rating:** 6
**Confidence:** 4

**Strengths:**

The addition of a github repo with example code significantly improves the presented benchmark's usefulness.  I have changed the score accordingly.

* The discussion of some of the nuances involved in NN quantization is useful.
* The comparison between the algorithms on multiple popular architectures is thorough and instructive.
* The authors' experiments showing the effects of different approaches to BatchNorm and graph structure are interesting.
* Overall the paper provides a good comparison of several quantization methods.


**Weaknesses:**

The weaknesses mostly pertain to use of the submitted work specifically as a benchmark.

* A good benchmark should be usable by future algorithms and methods.  If someone creates a new quantization algorithm, how should they use this benchmark?  There should be some clear instructions or code examples, which are absent.  Typically I would expect a git repository with code to execute the benchmark, instructions, and an example.

* It should be clear what the benchmark is measuring. Is the goal to measure how different quantization-aware training regimes perform when targeting a given inference-mode quantization scheme?  Is it meant to compare the accuracy or speed impacts of different inference-mode quantization schemes?  Something else?  Put another way, any benchmark fixes certain aspects of a problem, leaves other aspects of the problem as free design variables, and measures certain results.  It is not clear from the paper what is fixed, what is free, and what is measured.

* As a quantization overview, there should be some more discussion of the results.  The accuracy and accuracy degradation (FP accuracy - Quantized Accuracy) varies substantially from one model/setting to another, which the authors point out.  But are there some trends, groupings in the data, or underlying intuition to be taken from the results?  Do the data point to some guidance for an engineer choosing a quantization algorithm?

**Additional Feedback:**


* It seems like a missed opportunity that none of the targeted HW libraries support MCUs/DSPs.  Since edge inference on MCUs/DSPs is one of the biggest use cases for quantizing NNs, it seems like it would be useful to include something like TFLite Micro.

* L214/Table 4:  Can you discuss the differences in accuracy shown in Table 4?  Were the training configurations used to reproduce the results the same as those discussed in the paper?

* L303.  It appears that for the purpose of calculating difficulty, a setting/algorithm combination that does not converge counts as 0%, so that setting will have a "difficulty" equal to the highest accuracy for that setting.  This method makes the numeric difficulty not very meaningful and also not stable to new entrants.  For example, if another algorithm is entered into the benchmark that performs worse on a setting, the difficulty will increase.  This seems like an undesirable characteristic for a benchmark.


**Clarity:**

The paper is generally clearly written.  A few instances of unclear passages and mechanical issues are addressed here:

* Abstract. Several minor issues.  "By aligning up the training settings" should just be "By aligning the training settings." The phrase "about-the-same performance" should have no hyphens.   Line 15: "there is ... still a long way to go."  A long way to to what exactly?

* Line 41: "paper does not testify the algorithm " => "papers do not test the algorithm"

* L47: "Another noteless problem" Did you mean "notable?"

* L68: "evaluation is performed in 4 dimensions: supported hardware library, ..."  It becomes clear later on what is meant by HW library (a software library for performing NN tasks on given supported HW), but at this point in the paper it is not clear.  Please clarify the term.

* L96: The phrase "identify the quantizer type with learning-based which learns the scale and rule-based which directly computes" needs several corrections: "identify the quantizer type *as* learning-based, which learns the scale, *or* rule-based, which directly computes ..."

* L110: "As for 2-bit quantization, we find most of the algorithm will crash on hardware settings." Elaborate.  Did the QA training process fail to converge?  Did the training software crash?  With fake quantization, the quantities in the graph are still just FP numbers like in any other NN, so it is unclear why 2b quantization would cause a crash.

* Fig. 2(b) why is the value after ReLU ($x^{l+1}$) dequantized?  It is already an INT32.

* Section 3.2.  The discussion on batch-norm folding is confusing.  Folding is typically applied for inference, and computed after training, but several of the points discuss the the impact on training.  Is the benchmark meant to cover quantized training?

* It is not clear how the discussions of BatchNorm and block graph structure (3.2, 3.3) connect to the larger point of the paper.  These are two specific algorithm choices that are part of any quantization algorithm.  If the paper is intended as an overview of different techniques in NN quantization and their effects on accuracy, performance etc., then it makes sense to include these sections.  If the paper is describing a benchmark, then I would expect the focus to be on the benchmark and the included algorithms as they are published, without dissecting them.  If these block structures are design choices that a user needs to make after choosing target hardware and a quantization algorithm, then please clarify this fact.

* Table 7: At the bottom, footnotes 1,2,3,4 should be separated by commas.

* L326: "However, MQBench also has limitations, beyond image classification, quantization faces more challenges in deployments like object detection and NLP application." This sentence is unclear.



**Correctness:**

This reviewer did not find any major factual errors or methodological problems.  This work's weaknesses as a benchmark are discussed in the weaknesses section.

**Documentation:**

There is not suffficient detail to support reproducibility.  As discussed above, a benchmark of this type would typically have a repository with code to run the benchmark itself, instructions on how to apply the benchmark to new methods/algorithms, and an example.  All of these pieces are lacking.  This point is discussed in more detail in the weaknesses section.

**Ethics:**

I did not see any ethical concerns.

**Relation To Prior Work:**

Prior work is adequately addressed.

**Summary And Contributions:**

This paper discusses several algorithms for quantizing neural networks, compares the resulting accuracy, and identifies several challenges in quantization.

---

> ### Author Response · Authors · 2021-07-13
> **Response to Reviewer iQFC (Part 2)**
>
> A6: BN folding is applied for inference, however, during QAT we must fold BN prior to quantization to simulate the behavior in inference. This is because folding quantized weight does not equal quantizing the folded weight. Thus, folding cannot be applied after training. [1, 3] also emphasize the necessity of BN folding. We find there are different strategies to fold BN and in Sect. 3.3 we give a systematic analysis that is ignored in prior works. Note that  BN folding is for simulating quantized inference rather than quantized training.
>
> Q7: *It is unclear how the discussions of BatchNorm and block graph structure connect to the larger point of the paper. These are two specific algorithm choices that are part of any quantization algorithm. If the paper is intended as an overview of different techniques in NN quantization and their effects on accuracy, performance then it makes sense to include these sections. If the paper is describing a benchmark, I would expect the focus to be on the benchmark and the included algorithms as they are published, without dissecting them. If block structures are design choices that a user needs to make after choosing hardware and an algorithm, then please clarify this fact.*
>
> A7: We did not dissect existing algorithms. As we explained, existing algorithms use the wrong block graph and do not fold BN (see Table 2), thus they are not deployable in real-world cases. We think it would be meaningful to study how block graphs and BN folding affect the existing algorithms and how they contribute to a deployable benchmark. As can be seen in Table 7, deployable quantization typically has lower accuracy than academic quantization.
>
> Q8: *It seems like a missed opportunity that none of the targeted HW libraries support MCUs/DSPs. Since edge inference on MCUs/DSPs is one of the biggest use cases for quantizing NNs, it seems like it would be useful to include something like TFLite Micro.*
>
> A8: Thanks for the suggestion. MQBench already supports DSPs with Qualcomm's SNPE as shown in Table 2. MCUs are indeed not included in the current version. We check the documentation of TFLite Micro and find it supports a per-channel, asymmetric quantizer which is the same as the FBGEMM. It also supports a per-tensor, symmetric quantizer, a very similar configuration with TVM. The only difference is that TVM has a power-of-two scale, and TFLite Micro has an FP32 scale. We provide an example of how to extend MQBench hardware to TFLite Micro in our Github link. Due to the time limit, the experiments could not be finished. We intend to include more hardware and we'll add it in the future.
>
>
> Q9: *L214/Table 4: Can you discuss the differences in accuracy shown in Table 4? Were the training configurations used to reproduce the results the same as those discussed in the paper?*
>
> A9: The training configurations are not used to reproduce original results. As discussed in the introduction, some *old* algorithms have inferior accuracy because they do not use advanced training techniques. For example, finetuning from a pre-trained model always outperforms training from scratch. Thus, it is unclear whether the improvements of new algorithms come from the superiority of the algorithm rather than better training techniques. In Table 4, we reproduce all results with a better, unified pipeline, in order to compare quantization algorithms fairly.
>
>
> Q10: *L303. It appears that for the purpose of calculating difficulty, a setting/algorithm combination that does not converge counts as 0, so that setting will have a "difficulty" equal to the highest accuracy for that setting. This method makes the numeric difficulty not very meaningful and also not stable to new entrants. For example, if another algorithm is entered into the benchmark that performs worse on a setting, the difficulty will increase. This seems like an undesirable characteristic for a benchmark.}*
>
> A10: Thanks for your constructive advice. We agree with your opinion. This metric indeed has an undesirable characteristic. We switch to another metric, the standard deviation of the accuracy in each combination of network and hardware. As shown in Fig. 5, this variance study shows the sensitivity to quantization. Some cases are not sensitive, and algorithms are low-variance. While some are sensitive, corresponding to high variance. This also exhibits the diverse nature of MQBench.
>
> Q11: *" However, MQBench also has limitations, beyond image classification, quantization faces more challenges in deployments like object detection and NLP application." This sentence is unclear.*
>
> A11: In this sentence, we discuss the limitation of MQBench. More tasks are needed to evaluate quantization. However, due to the lack of baselines, we did not try other tasks for now. We hope this can be explored in future work. Sorry for the incomplete sentence.
>
> Q12: Other grammar errors
>
> A12: Thanks, we have revised them.

---

> ### Author Response · Authors · 2021-07-13
> **Response to Reviewer iQFC (Part 1)**
>
> Thanks for the advice and feedback. Below is our detailed reply, please reply to us if you have further questions.
>
> Q1: *If someone creates a new algorithm, how should they use this benchmark? There should be some clear instructions. Typically I would expect a git repository with code to execute the benchmark, instructions, and an example*
>
> A1: Thanks for your constructive advice, please see our GitHub repository for the instructions on how to reproduce MQBench. We also add a step-by-step guide for researchers who intend to benchmark their newly designed quantization algorithms. A code example for LSQ is provided. Besides, in Section 4 we add a brief description of our implementation.
>
> Q2: *Is the goal to measure how different QAT regimes perform when targeting a given inference-mode scheme? Is it meant to compare the accuracy or speed impacts of different inference-mode quantization schemes? Something else? It is not clear from the paper what is fixed, what is free, and what is measured.*
>
> A2: In this work, we mainly study the reproducibility and deployability problems of quantization **algorithms**, including PTQ (appendix) and QAT. Existing work runs on an academic setting that does not fold BN layers and has the wrong block graph for activation quantization. Thus they are undeployable in the practical hardware. Also, the different training hyper-parameters adopted prevent fair comparison. In MQBench, we first study the reproducibility's impact by unified training hyper-parameters in Sect. 5.1. Then, we study two essential deployability factors (BN folding and block graph) in Sect. 5.2 \& Sect. 5.3. Finally, we combine them and give a major benchmark in Sect. 5.4. All these efforts help identify useful algorithms under the practical scene. We apologize for not stating this explicitly.
>
> "*Is it meant to compare the accuracy or speed impacts of different inference-mode quantization schemes?*" We do not intend to compare different inference libraries and just want to evaluate algorithms comprehensively under multiple inference libraries.
>
> Q3: *As an overview, there should be some more discussion of the results. Are there some trends, groupings in the data, or underlying intuition to be taken from the results? Do the data point to some guidance for an engineer choosing a quantization algorithm?*
>
> A3: Thanks for this valuable advice. In our original paper, we include the discussion of data groupings and new findings in Sect 5.3 and 5.4 such as several insights for BN folding and block graphs. In Appendix B, we summarize according to some diagnostic information. In our revision, we also add more analysis of underlying rules in Section 5.4. In Fig. 6, we measure the mean accuracy of algorithms on different network architectures and hardware. As for the algorithm selection for quantization engineers, we add a new paragraph to provide instructions. Specifically, if the target hardware uses per-channel quantization, we recommend using PACT. In the case of per-tensor quantization, we recommend LSQ. Please check the paper for more details, thank you.
>
> Q4: *"As for 2-bit quantization, we find most of the algorithm will crash on hardware settings." Elaborate. Did the QA training process fail to converge? Did the training software crash? With fake quantization, the quantities in the graph are still just FP numbers like in any other NN, so it is unclear why 2b quantization would cause a crash?*
>
> A4:  In Table 7 you can find 4-bit quantization on real-world hardware could be very challenging (several cases fail to converge). We find 2-bit deployable quantization of both weights and activations causes a larger gap. For example, 2-bit symmetric quantization of activation is equal to 1-bit quantization for unsigned activations since the zero point is fixed to 0. And folding BN further increases the instability in training. As a result, the 2-bit network easily fails to converge. The crash is not software problems like NaN or overflow. Sorry for the lack of details of this.
>
> Q5: *Fig. 2(b) why is the value after ReLU () dequantized? It is already an INT32*.
>
> A5: Mathematically, the requantization operation equal to a dequantization and a quantization operation. However, in real world hardware, the Requantization operation is not performed using such two-stage form but a single operation. We can rewrite Eq.3 to
> $$\bar{\mathbf{x}}^{l+1} = \mathrm{clip}(\left\lfloor \frac{{\mathbf{x}}^{l+1}}{s_{\hat{{x}}^{l+1}}}\cdot s_{w^l}\cdot s_{x^l} \right\rceil+z_{\hat{{x}}^{l+1}},N_{min}, N_{max})$$
> Since the de-quantization operation does not exist in practice, we do not draw the format of the dequantization output in Fig. 3. You can also find the details of requantization in [3].
>
> Q6: *Section 3.2. The discussion on batch-norm folding is confusing. Folding is typically applied for inference and computed after training, but several of the points discuss the impact on training. Is the benchmark meant to cover quantized training?*

---

### Author Response · Authors · 2021-07-13
**General Response**

We thank all reviewers for their efforts and feedback on MQBench. All reviewers raise the concern of lacking the documentation, code, website for a benchmark. We apologize for that. We have been doing our best to improve it since the submission, and also revised our submission manuscript. We hope our revision addresses the concerns. Thank you.

Our revision includes:

1. Benchmark website (http://mqbench.tech/), we add leaderboard, submission instructions.
2. Framework GitHub (https://github.com/TheGreatCold/mqbench), we add a detailed README.md to reproduce our results and instructions to add new algorithms & hardware.
3. Documentation of MQBench (http://mqbench.tech/assets/docs/html/), the documentation of MQBench.
4. Paper revision:

    4.1 In L204, we add a new section discussing our implementation of MQBench.

    4.2 In Section 5.4 we add more visualization results to analyze the underlying rules of algorithms.

---

### Decision · Program_Chairs · 2021-07-26

**Decision:**

Accept

**Comment:**

The paper introduces a benchmark for model quantization. Reviewers found the paper to be clear and the benchmark useful, but raised concerns primarily about the reproducibility and extensibility of the benchmark. These limitations were mostly addressed through the authors’ response and revised paper, which also include a link to open-source code which makes the proposed benchmark significantly more useful. Congratulations on having your paper accepted to the NeurIPS 2021 Track on Datasets and Benchmarks! The authors are encouraged to take the feedback from reviewers into account when preparing the final version of their paper.